# Caloric restriction leads to druggable LSD1-dependent cancer stem cells expansion

Rani Pallavi [1], Elena Gatti[1,10], Tiphanie Durfort [1,10], Massimo Stendardo[1], Roberto Ravasio [1], Tommaso Leonardi [2], Paolo Falvo[1], Bruno Achutti Duso [1], Simona Punzi[1], Aobuli Xieraili[1], Andrea Polazzi[1], Doriana Verrelli [1], Deborah Trastulli [1], Simona Ronzoni[1], Simone Frascolla[1], Giulia Perticari [1], Mohamed Elgendy[1,3,4,5,6], Mario Varasi[7], Emanuela Colombo [1,8], Marco Giorgio[1,9], Luisa Lanfrancone [1], Saverio Minucci[1,8], Luca Mazzarella [1] ✉ & Pier Giuseppe Pelicci [1,8] ✉

Caloric Restriction (CR) has established anti-cancer effects, but its clinical relevance and molecular mechanism remain largely undefined. Here, we investigate CR's impact on several mouse models of Acute Myeloid Leukemias, including Acute Promyelocytic Leukemia, a subtype strongly affected by obesity. After an initial marked anti-tumor effect, lethal disease invariably re-emerges. Initially, CR leads to cell-cycle restriction, apoptosis, and inhibition of TOR and insulin/IGF1 signaling. The relapse, instead, is associated with the non-genetic selection of Leukemia Initiating Cells and the downregulation of double-stranded RNA (dsRNA) sensing and Interferon (IFN) signaling genes. The CR-induced adaptive phenotype is highly sensitive to pharmacological or genetic ablation of LSD1, a lysine demethylase regulating both stem cells and dsRNA/ IFN signaling. CR + LSD1 inhibition leads to the re-activation of dsRNA/ IFN signaling, massive RNASEL-dependent apoptosis, and complete leukemia eradication in ~90% of mice. Importantly, CR-LSD1 interaction can be modeled in vivo and in vitro by combining LSD1 ablation with pharmacological inhibitors of insulin/IGF1 or dual PI3K/MEK blockade. Mechanistically, insulin/IGF1 inhibition sensitizes blasts to LSD1-induced death by inhibiting the anti-apoptotic factor CFLAR. CR and LSD1 inhibition also synergize in patient-derived AML and triple-negative breast cancer xenografts. Our data provide a rationale for epi-metabolic pharmacologic combinations across multiple tumors.

Accumulating clinical and preclinical evidence supports the role of caloric restriction (CR) or other dietary-restriction interventions in controlling tumor growth and response to anti-cancer therapies[1–3], though the response of different tumor types varies widely[4,5]. CR reduces nutrient availability and circulating levels of insulin and insulin-like growth factor (IGF1)[6–8], and induces a wide range of effects on both normal and tumoral cells, including reduced mTOR activation[9,10], activation of sirtuins[11], reduced protein synthesis[12], increased autophagy[13,14], reduced inflammation[15], changes in stem cells[16] and immunity[17]. Our understanding of such complex interactions at the molecular level remains far from complete, as reductionist in vivo or in vitro approaches can only model one component at a time, possibly removing the effects of the others. Furthermore, the effect of CR on Cancer Stem Cells (CSCs), or Leukemia Stem Cells (LSCs) in leukemias, is poorly understood.

No systematic studies have yet clearly identified tumor types with significantly higher response rates to CR or other dietary interventions. However, we can indirectly rely on a large body of evidence on the relationship between adiposity (body mass index) and tumor outcome to identify tumors that are likely to respond to dietary modulation. Among hematological malignancies, we showed that Acute Promyelocytic Leukemia (APL), a subtype of Acute Myeloid Leukemia (AML), almost invariably characterized by t (15;17) translocations coding for the PML/RARa fusion protein, is strongly negatively-affected by an elevated body mass index[18,19]. Established syngeneic APL mouse models allow leukemia transplantation in immunocompetent, non-irradiated hosts, features that provide a clinically relevant and experimentally tractable system to investigate mechanistic changes imparted by CR on cancer cells without confoundings associated with immunodeficiency or radiation-associated wasting syndrome.

Here, by exploring CR-induced phenotypic and molecular alterations in mouse APL, we identify a paradoxical LIC-promoting effect that crucially depends on IGF1/insulin signaling and the double-stranded RNA-suppressing activity of the lysine demethylase LSD1. We show that co-targeting IGF1/insulin and LSD1 leads to significant anti-tumoral response not only in APL but also in other AML subtypes and triple-negative breast cancers, suggesting a therapeutic strategy.

## Results

### CR paradoxically increases Leukemia Initiating Cells

To analyze the anti-leukemic effects of CR on AMLs, we used mouse models representative of its most frequent genetic subtype (NPM1c+, NPM1c+FLT3-ITD, MLL-AF9, and PML/RARa)[20–23]. Mice were subjected to CR starting at 2 weeks before injection of CD45.2 + AML blasts into congenic (CD45.1+) mice[20]. CR significantly prolonged survival of mice injected with AMLs expressing PML/RARa[21], NPM1c+[22], and NPM1c+FLT3-ITD[22], but not in MLL-AF9 AMLs as previously reported[23] (Supplementary Fig. 1a–d). Given prior evidence on the impact of obesity on PML-RAR AMLs[19], we expanded our investigation on the APL model.

CR-fed mice showed significantly increased survival over mice fed with *ad libitum* standard diet (SD), irrespective of whether CR was initiated 2 weeks prior to leukemia injection, at the same time or 4 weeks after (Fig. 1a). The survival advantage was preserved in immunodeficient mice (*Rag*-deficient mice - RAG−/−, or NOD.Cg-*Prkdc^scid^ Il2rg^tm1Wjl^*/SzJ mice - NSG) (Supplementary Fig. 1e) suggesting that the CR-increased survival is not due to clearance of leukemia cells by immune cells (T and B cells; natural killer cells). In subsequent experiments, CR was administered at 2 weeks prior to leukemia injection unless otherwise specified.

CR led to a significant delay in blast accumulation in the peripheral blood (PB), bone marrow (BM), and spleen (Fig. 1b–d and Supplementary Fig. 1f–i). CR blasts were less proliferative (by Ki67 and DAPI; Fig. 1e) and showed a modest trend for higher apoptosis (by cleaved caspase-3 positive) (Supplementary Fig. 1j–l). These anti-tumor effects confirm previous reports of CR in various tumor-types[24–26]. However, leukemia ultimately developed in all CR mice, which at 6 weeks (when CR mice showed no clinical signs of disease, while ~1/3 of the SD mice were already dead) showed levels of blast infiltration in the BM and spleen comparable to SD mice (Fig. 1c, d). Notably, blast re-expansion was observed in all CR-treated mice, which eventually succumbed to the disease (Fig. 1a). Thus, CR induces a transient anti-leukemic response, which is invariably followed by lethal blast-expansion.

As leukemia progression and relapse are thought to be sustained by a rare subset of LSCs (experimentally defined as "Leukemia Initiating Cells", LICs), we measured LIC frequency in APL blasts from SD- or CR-fed mice, by injecting CD45.1+ mouse recipients with decreasing amounts of CD45.2 + FACS-purified blasts (extreme limiting-dilution

transplantation analyses – ELDA)[27]. Cells were collected at 4 weeks after injection, e.g., prior to the re-expansion of CR-conditioned blasts. Surprisingly, CR-conditioned APLs showed significantly increased LIC content (~15-fold, $p = 3.9e−05$), with a frequency of 1/497 vs. 1/7,751 in the SD-APLs (Fig. 1f and Supplementary Fig. 1m). Consistently, we found ~2.5-fold enrichment upon CR of a cell population defined as CD34+, c-Kit+, FcγRIII/II+, Gr1^int ("CKFG" cells), and previously shown to be ~100 fold enriched in LIC activity in the blasts from PML-RAR transgenics[28] (Fig. 1g and Supplementary Fig. 1n). Similarly, CR led to a ~3.5-fold increase in the CKFG population in the NPM1c+ AMLs (from ~0.09% in SD to 0.32% in CR (Fig. 1h). Upon secondary transplantation in limiting dilution, a non-statistically significant trend for a faster leukemia development could also be appreciated in recipients of CR-conditioned leukemia (Supplementary Fig. 1o–r).

Thus, in multiple models of CR-responsive AMLs, the transient anti-leukemic response is invariably followed by the expansion of cells enriched for LIC activity and disease relapse. This is reminiscent of the reported stimulatory effect of CR on self-renewal of hematopoietic stem cells (HSCs)[29], and our observation of increased numbers of long-term HSCs upon CR (LT-HSCs: Lineage−, Sca+, c-Kit+, CD34−, Flk−; Supplementary Fig. 4f), suggesting that CR activates in leukemic blasts a conserved adaptive response to energy deprivation.

### CR downregulates endogenous dsRNAs and Interferon signaling

To investigate molecular mechanisms underlying the effect of CR on LICs, we analyzed the genomic and transcriptional landscapes of treated PML-RAR – expressing leukemias. Whole Exome Sequencing (WES) ruled out genetic selection as a mechanism for LIC expansion since leukemias grown in SD and CR recipients exhibited a small number of acquired (i.e., not detected in the common leukemic population inoculated in all mice) mutations (4 in CR vs. 3 in SD; $p > 0.1$; Supplementary Fig. 2a). Most notably, all mutations were subclonal and non-recurrent, and did not affect significantly the putative function of the affected gene, nor involved known cancer drivers (Supplementary Fig. 2b, c and Supplementary Data 1).

RNA sequencing, instead, revealed dramatic transcriptional rewiring of CR-conditioned cells, with significant modulation of the expression of 1160 genes (642 downregulated and 518 upregulated genes; Fig. 2a). Gene Set Enrichment Analyses (GSEA) showed significant downregulation of gene sets known to mediate the effects of CR on life span and tumor growth[9,10,30–33], such as the Myc signaling pathways, and targets of insulin/IGF1 signaling (Fig. 2b), including the immediate-early response genes *Fos, Jun, Egr1, Egr2, Nr4a1, Atf3, Socs3* (Supplementary Fig. 3a), consistent with the reduced circulating levels of IGF1 in CR-fed APL mice[6,7] (Supplementary Fig. 3b). Other pathways or genes previously implicated in the effect of CR were not affected, including the leptin receptor[34], heme-oxygenase-1[35] or known pathways involved in normal hematopoietic SC maintenance[29].

The two most-depleted signaling pathways in CR-APLs were interferon (IFN) alpha and gamma (Type I IFN) (FDR < 0.0001, NES = −2.07 and FDR < 0.0001, NES = −2.08 respectively; Fig. 2b, c). Inspection of individual genes showed marked down-regulation of several IFN-stimulated genes (ISGs) (*Irf1, Irf7, Irf9, Stat1, Stat2, Isg15, Isg20, Ifit1*, MHC class I genes - *H2-K1, H2-D1, H2-Q5, H2-T22, H2-T23,* and *H2-M3*), and *B2m* (Fig. 2d). Most marked downregulation was observed for factors involved in the cytoplasmic recognition of double-stranded RNA (dsRNA), including the OAS oligoadenylate-synthases (*Oas1a; Oas2,3; Oasl1-2*), the *Rig-I, Mda5,* and *Lgp2* (RIG-I-like) helicases (Fig. 2d)[36–39]. As dsRNA[40] may derive from endogenous repetitive elements such as ERV/LTRs, SINE, and LINE family members, we specifically quantified the expression of these elements using a cluster-based approach, and found a generalized downregulation of most clusters (Fig. 2e).

Together, these data suggest that a major component of the transcriptional reprogramming induced by CR is the downregulation

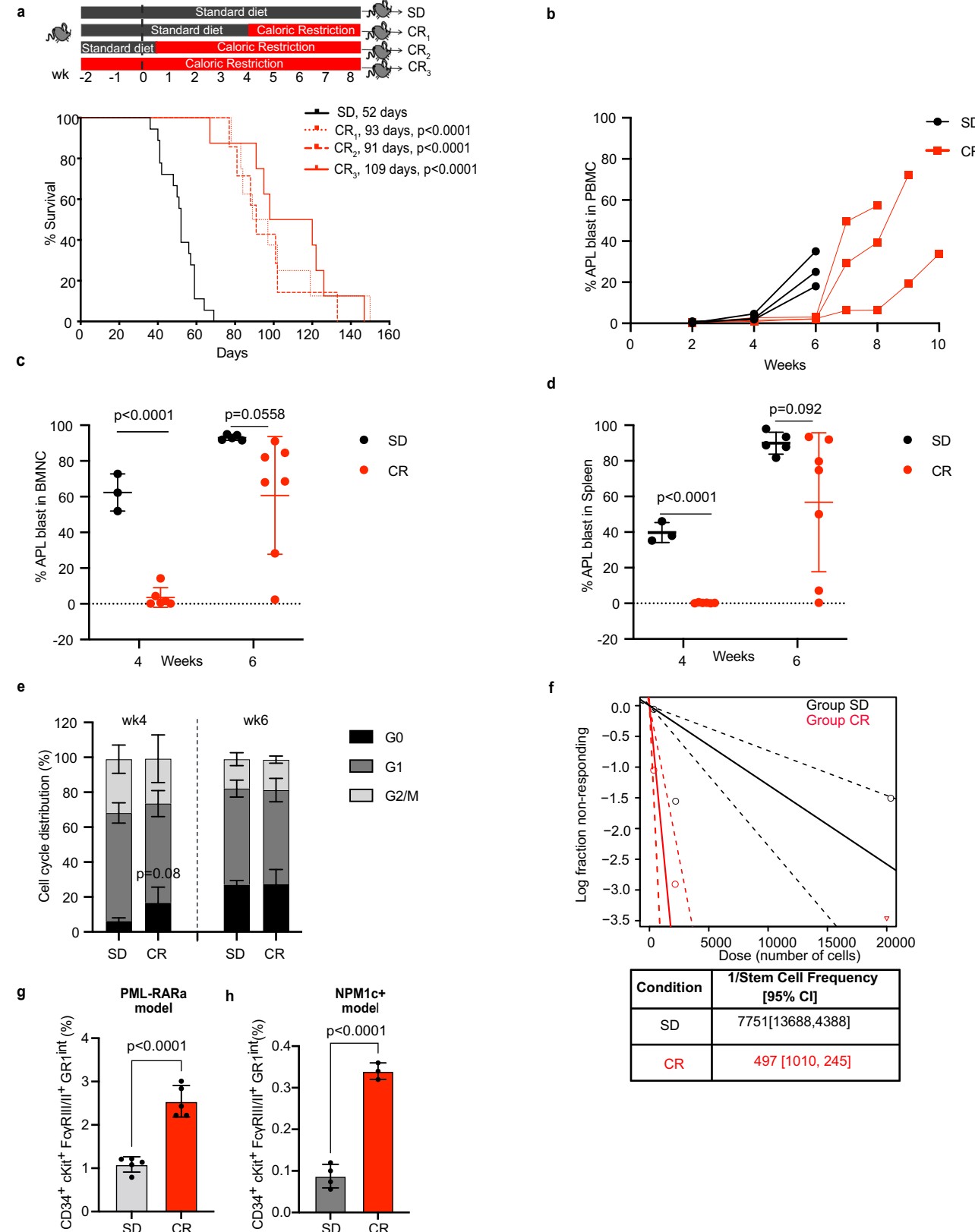

of dsRNA formation and attenuation of downstream IFN signaling, in the absence of activation of specific SC-programs. Interferons are potent inducers of normal and leukemic SC-exhaustion[41,42], suggesting that decreased IFN signaling may preserve LIC function upon CR, leading to the observed LIC increase.

## LSD1 inhibition eradicates leukemia upon CR

We hypothesized that reduced dsRNA/IFN signaling upon CR may provide a target for synergistic therapeutic approaches. IFN signaling from endogenous dsRNA is regulated in cancer cells at the chromatin level by the histone demethylase LSD1[38], which has long been known to

**Fig. 1 | CR induces transient inhibition of leukemia growth but increases numbers of CR-adapted Leukemia Initiating Cells. a** Schematic representation (upper panel) and Kaplan–Meier survival curve (lower panel) of mice injected with 2 ×10$^6$ APL cells and randomized to standard diet (SD, $n = 18$) or caloric restriction (CR) initiated 4 weeks after injection (CR$_1$, $n = 8$), 2 days after injection (CR$_2$, $n = 7$) or 2 weeks before injection (CR$_3$, $n = 8$). All CR-fed mice showed a survival advantage with a median survival of 93 days, $p < 0.0001$ in CR$_1$; 91 days, $p < 0.0001$ in CR$_2$ and 109 days in CR$_3$, $p < 0.0001$, as compared to 52 days in SD. Statistical analysis was performed using log-rank (Mantel–Cox) test. **b** Kinetics of APL blasts (CD45.2+) in the peripheral blood of individual recipient CD45.1+ mice subjected to SD or CR$_3$ ($n = 3$ mice per group). APL blasts (CD45.2+) in the BM (**c**) or spleen (**d**) of recipient CD45.1+ mice subjected to SD or CR$_3$, 4 (SD, $n = 3$; CR, $n = 6$) or 6 (SD, $n = 5$; CR, $n = 7$) weeks after APL injection. **e** Cell cycle analysis of APL blasts in the BM at 4 (SD, $n = 4$; CR, $n = 4$) and 6 (SD, $n = 4$; CR, $n = 3$) weeks post-injection. **f** A log fraction plot of limiting dilution model fitted to the data (Supplementary Fig. 1e) by ELDA. Dotted line represents the 95% of confidence interval (upper). Table reporting the LIC frequency with confidence intervals (lower). Relative frequency of the CD34$^+$, c-Kit$^+$, FcγRIII/II$^+$, Gr1$^{int}$ subpopulation in SD and CR APLs ($p < 0.0001$, SD, $n = 5$; CR, $n = 5$) (**g**) and SD and CR NPM1c+ AMLs ($p < 0.0001$, SD, $n = 4$; CR, $n = 3$) (**h**). Data are expressed as mean ± SD for **c–e**, **g** and **h**. Statistical analysis was performed using two-tailed t-test for **c–e**, **g** and **h**. Here 'n' represents number of mice/groups. Source data are provided as a Source Data file.

be required for LIC maintenance in some AML models[43]. Notably, though LSD1 mRNA levels were unaffected by CR (Supplementary Fig. 4a; upper panel), levels of LSD1 protein were markedly upregulated in APL blasts upon CR (Fig. 3a; Supplementary Fig. 4a; lower panel).

LSD1 is the object of intensive drug development efforts, with LSD1 inhibitors currently in clinical or pre-clinical development for the treatment of hematological neoplasms[44]. Thus, we treated SD- or CR-fed APL-leukemic mice with our in-house generated LSD1 inhibitor (LSD1i)[45], starting 3 weeks after leukemia injection. LSD1i in SD mice was partially effective, leading to prolonged survival in 2/5 mice (>300 days with no signs of leukemia at the sacrifice; $p < 0.0001$). Strikingly, when LSD1i was administered to CR mice, it led to normalization of the bone marrow and spleen architecture (Supplementary Fig. 4b–d), and prolonged survival in 8/9 mice (Fig. 3b). Analyses of the BM shortly after LSD1i treatment (+1 week) revealed a dramatic reduction (-10$^4$) or even complete loss of leukemic blasts in CR mice, quantitatively superior to that induced in SD mice (Fig. 3c), accompanied by significantly increased apoptosis (Fig. 3d). Again, the effects of LSD1i in SD were significantly-less marked (Fig. 3 c, d).

To obtain formal proof of complete disease eradication, we transplanted total BM cells (-8 × 10^6 cells *per* recipient) from long-surviving mice into RAG−/− mice ($n = 5$ *per* donor). Death from leukemia was observed in recipients of BM from only 1/5 CR+ LSD1i -treated mice vs. 1/2 SD+ LSD1i mice (Supplementary Fig. 4e). To obtain genetic confirmation of the synergism between CR and LSD1 inhibition, we ablated the LSD1 gene in the NB4 APL cell line by CRISPR (LSD1-KO cells). NSG mice were injected subcutaneously with WT or LSD1-KO cells and fed either SD or CR. There was no difference in local growth in the two SD-fed cohorts. As expected, the growth of WT cells was significantly reduced in CR- vs. SD-fed mice. The growth of KO cells was instead almost completely abrogated in the CR-fed mice (Fig. 3e).

Thus, LSD1 inhibition eradicates leukemias upon CR, suggesting that the adaptive response of LICs to energy deprivation is absolutely required for their survival.

Treatment with the LSD1i did not reduce numbers of LT-HSCs in CR-fed mice, nor it did it in most other BM subpopulations, including multilineage progenitors (MPPs; Lineage−, Sca+, c-Kit+, CD34+, Flk+), myeloblasts (which were increased), myeloid derived suppressor cells (no change), myelocytes (no change), with the exception of erythroid-committed cells, which are well-known specific targets of LSD1i[44], causing a reversible anemia (Supplementary Fig. 4f), thus suggesting that the LSD1-dependency is a specific vulnerability of leukemic cells.

## LSD1 inhibition synergizes with insulin/IGF1r inhibition

We then investigated which of the main CR-targeted signaling pathways, mTOR or insulin/IGF1, synergize with LSD1 inhibition. LSD1i was administered together with the CR-mimetic drugs OSI-906 or rapamycin, which inhibit insulin/IGF1 receptor or mTOR, respectively[13]. Treatment of APL-bearing SD-fed mice with LSD1i or OSI-906 alone had a modest yet significant effect on survival, while treatment with rapamycin exerted no effect (Fig. 3f). Concomitant LSD1i - rapamycin treatment showed no synergistic effect, suggesting that mTOR inhibition is not sufficient to sensitize leukemia cells to LSD1 inhibition

in vivo, although we cannot rule out suboptimal dosing despite usage of a relatively-high dosage[45]. Concomitant LSD1i -OSI-906 treatment, instead, led to significantly-increased survival compared to each drug alone, suggesting that LSD1 inhibition synergizes with attenuation of insulin/IGF1R signaling (Fig. 3f).

Notably, treatment of NB4 APL cells in vitro with OSI-906 induced LSD1 upregulation (Supplementary Fig. 4g), reduced cell growth in a dose-dependent manner (Fig. 3g), while it increased significantly the frequency of colony-forming cells, even at the highest concentrations (Fig. 3h), an effect that is reminiscent of that of CR on APL in vivo (e.g., reduced cell proliferation and increased LIC frequency). Treatment with LSD1i did not modify significantly the frequency of colony-forming cells when used alone, while it abrogated completely the effect of OSI-906 on colony formation (Fig. 3h), thus providing a reductionist in vitro system for further mechanistic studies.

## Co-inhibiting LSD1 and insulin/IGF1R sensitizes leukemia to TRAIL-induced death

RNAseq analyses of 4-week CR-conditioned and LSD1i-treated APLs showed dramatic transcriptional rewiring, as compared to CR-conditioned APLs, with significant modulation of the expression of 3302 genes (2190 upregulated and 1112 down-regulated genes; Fig. 4a). Most notably, LSD1i reverted the effects of CR on dsRNA/IFN signaling completely (Fig. 4b), with dramatic upregulation of dsRNA sensors and other ISGs (Fig. 4c)[46]. Consistently, LSD1i elicited de-repression of dsRNA-inducing ERVs, LINE, and SINE (Supplementary Fig. 5a), in a way that was linearly and inversely correlated with CR-induced down-regulation (Fig. 4d), suggesting that LSD1 and CR regulate the same set of transposable-elements (TE) -derived RNA sequences. Western blotting confirmed increased expression of the dsRNA sensors RIG1 and OAS1 and phosphorylated PKR (Supplementary Fig. 5b), a stress-responsive kinase that is specifically activated by dsRNA[47]. Upregulation of ISGs was observed irrespective of concomitant CR or OSI-906 (Fig. 4b, c; Supplementary Fig. 5c). Transcriptional differences were significantly attenuated or inverted in leukemias isolated at later time points (6 weeks, Supplementary Fig. 5d), when organ infiltration is advanced and signs of systemic distress start appearing.

To increase granularity of the biological changes induced by CR and LSD1 inhibition and their effects on LICs, we performed single-cell RNAseq (scRNAseq) on purified SD/CR APLs ± LSD1i, each in duplicate. In total we obtained high quality profiles for 18,020 cells (range 4000–4500 cells per condition, SD = 4529; CR = 4015; SD + LSD1i = 4917; CR+LSD1i = 4559), with high consistency between biological duplicates (Supplementary Fig. 6a). Hierarchical clustering identified an optimal set of 19 clusters of cells sharing similar gene-expression patterns, which accounted for 95.6% of the total variance in the dataset (Fig. 4e; Supplementary Fig. 6b). Analyses of the cluster phylogenetic-tree revealed a first bifurcation involving a set of 3 clusters (15, 16 and 17), characterized by increased expression of *Cd177* (Fig. 4f; Supplementary Fig. 6c), recently identified as a negative prognostic factor for myeloid leukemias[48].

To map putative LICs in the scRNAseq dataset, we analyzed expression of the CKFG markers. Of these, only *Cd34* appeared

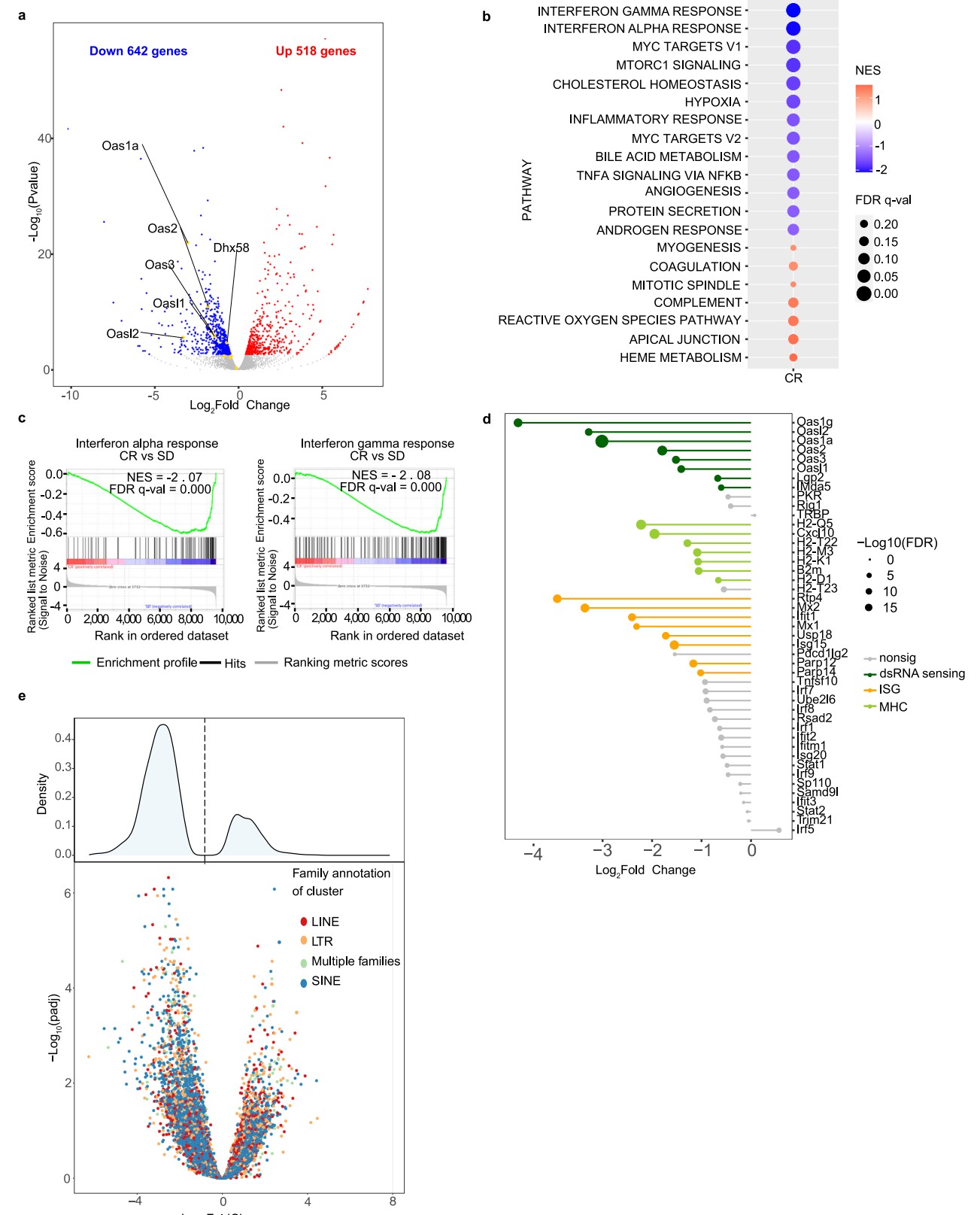

informative, since the others were either undetectable (*c-kit* and *Ly6g* (*Gr1*)) or uniformly expressed (*Fcgr2b*) across clusters (Supplementary Fig. 6c). Combining *Cd34* and *Cd177* identified a double-positive population (scLICs) with frequency increasing from 0.55% in SD to 2.26% in CR (Fig. 4g). CD177 positivity was confirmed by FACS in the putative LIC population (now defined as CKFG/CD177hi) in both APLs

and NPM1c+ leukemias (Supplementary Fig. 6d). Similar to bulk RNA-seq, scRNAseq profiles of samples obtained at later time points did not clearly identify a separate cell population (Supplementary Fig. 6e, f).

scLICs showed properties of a metabolically quiescent pheno-type, previously associated with dormant HSCs[49,50], including decreased expression of OXPHOS, MYC, proliferation, and Interferon

**Fig. 2 | Caloric restriction decreases the IFN/dsRNA signaling in the APL blasts.** **a** Volcano plot showing differentially expressed genes in CR and SD APL cells (RNAseq analyses from a pool of 2 mice per condition). Genes in red or blue are significantly up or down-regulated (FDR < 0.05). dsRNA sensing genes of interest are labeled. **b** hallmark GSEA terms enriched/depleted in CR and SD. The color of the dots indicates the normalized enrichment score (NES). The size indicates the FDR *q*-value. Note that only dots corresponding to pathways with an FDR < 0.25 are reported. **c** GSEA plots showing enrichment/depletion in CR vs SD of the interferon alpha and interferon gamma signatures, as indicated. **d** Differential expression of dsRNA sensing genes, MHC genes and other ISGs between CR and SD. Genes in gray("nonsig") are not significantly differentially expressed, with an FDR > 0.01. **e** Differential expression of ERV families from stranded RNAseq in CR *vs.* SD. Upper panel: density plot of log2 fold change; bottom panel: volcano plot of log2 fold change vs -log10 FDR. Colored dots are significantly deregulated genes at FDR < 0.05. Source data are provided as a Source Data file.

alpha/gamma gene sets, maintained in all treatment groups (Fig. 4h and Supplementary Fig. 6g). scLIC cells also exhibited coexisting low-intermediate levels of the transcription factors *Irf8* and *Gfi1* (Fig. 4i), which in normal bone marrow identify a rare, phenotypically unstable progenitor population that rapidly commits to either Gfi1^high^/Irf8^- granulocytes or Gfi1^-/Irf8^high^ monocytes[51]. Upon LSD1i, however, most cells, including scLICs, upregulated *Irf8* expression (Supplementary Fig. 6h), as observed in bulk RNAseq data, and became *Gfi1*^high^/*Irf8*^high^, a bi-differentiated cell state presumably not compatible with cell survival (Fig. 4i). Notably, *Irf8* was recently identified as a key factor for the development of PML-RAR expressing AMLs (APLs)[52] and sensitizes myeloid cells to extrinsic, TNF/TRAIL-induced apoptosis[53]. Consistently, a signature predictive of TRAIL sensitivity[54] was significantly upregulated upon LSD1i treatment (Fig. 4j). Thus, LSD1 inhibition, through the activation of an interferon-associated transcriptional program, induces a phenotypically unstable and apoptosis-prone state in APL cells, including LSCs.

To directly test if APL cells are sensitized to apoptosis by LSD1i, we assessed the effect of TRAIL on NB4 cells upon attenuated insulin/IGF1 signaling, using OSI-906, and/or pharmacological or genetic ablation of LSD1 (using LSD1i or LSD1-KO NB4 cells, respectively). In the absence of TRAIL, perturbation of IGF1/insulin or LSD1 signaling, alone or in combination, had no major effect on cell survival. Upon TRAIL addition, the combination of OSI-906 with pharmacological or genetic loss of LSD1 markedly increased apoptosis up to ~3-fold (Fig. 5a, b; Supplementary Fig 7a). Similar results were obtained in NPM1c+ leukemia cells exposed to OSI-906 + LSD1i (Fig. 5c). Notably, TRAIL sensitization did not require paracrine IFN secretion, since RNAseq revealed no expression of IFN transcripts in NB4 cells, and could be fully recapitulated by LSD1 ablation in the presence of combined pharmacological inhibition of PI3K (using Ly294002) and MEK (using trametinib), the two main downstream effectors of insulin/IGF1, but not by either inhibitor alone (Fig. 5d; Supplementary Fig. 7b), suggesting that all branches of the insulin/IGF1 signaling pathway must be attenuated in order to synergize with LSD1 inhibition.

We then investigated whether sensitization to extrinsic apoptosis depends on activation of the dsRNA-sensing machinery, by silencing the critical mediator RNASEL[38,55], using short hairpin RNA (shRNA) in WT or LSD1-KO NB4 cells (Supplementary Fig. 8a). RNAseL depletion abolished the sensitivity of NB4 LSDKO + OSI-906 cells to TRAIL-induced apoptosis (Fig. 5e and Supplementary Fig. 8b).

## CFLAR mediates CR-induced sensitization to apoptosis

We then investigated why LSD1i-induced apoptosis depends so critically on the concomitant blockade of the insulin/IGF1 pathway. TRAIL triggers extrinsic apoptosis through FADD-dependent activation of caspase-8, −10, −3[56] and the DISC death-inducing signaling complex. Consistently, OSI-906 markedly increased TRAIL-induced caspase3 activation in LSD1-KO NB4 cells (Supplementary Fig 8c, d). Notably, cell death was dependent on both caspase-3 and −8 activation, as pan-caspase (Z-VAD-FMK), caspase-3 (Z-DEVD-FMK) or −8 (Z-IETD-FMK) inhibitors were all equally capable to rescue TRAIL-induced death in OSI-906 treated LSD1-KO NB4 cells (Fig. 5f). The CFLAR (CASP8 and FADD Like Apoptosis Regulator) competes with procaspase-8 in the formation of DISC and is a potent inhibitor of TRAIL-induced

apoptosis[57]. Notably, CFLAR is a key point of metabolic control of apoptosis, since it requires de novo protein synthesis[58], and is down-regulated under limited nutrient conditions[59]. CFLAR was down-regulated in mouse APL blasts upon CR in vivo (Fig. 5g, h). In NB4, LSD1-KO NB4 and NPM1c+ cells in vitro, treatment with OSI-906 led to reduced CFLAR levels in response to TRAIL stimulation (Fig. 5i-j and Supplementary Fig. 8e). We also noticed that pharmacological LSD1 ablation per se leads to some degree of CFLAR downregulation, which may further contribute to the pharmacological synergy (Fig. 5i).

CFLAR down-regulation likely occurs due to a generalized reduction in transcription and translation rate associated with lower insulin signaling, which disproportionately affects CFLAR, a high turnover protein upregulated de novo upon apoptotic stimuli[59]. Indeed, Insulin/IGF1 blockade led, as expected, to significant decrease in de novo overall translation (Supplementary Fig. 8f). Direct inhibition of translation or transcription using non-lethal doses of actinomycin D or cycloheximide led to similar reduction in baseline and TRAIL-induced CFLAR (Supplementary Fig. 8g) and to TRAIL-induced apoptosis in LSD1-KO cells (Supplementary Fig. 8h, i).

To confirm the role CFLAR in OSI-906 - dependent apoptosis, we overexpressed CFLARs in NB4 and LSDKO cells (Supplementary Fig. 8j). Importantly, transgenic CFLAR levels were not affected by OSI-906 (Supplementary Fig. 8k). Overexpression of CFLARs led to generally increased survival, but most importantly completely abrogated TRAIL-induced death in OSI-906-treated LSD1-KO cells, as shown by both PI/annexin V and cleaved caspase-3 quantification (Fig. 5k, l and Supplementary Fig. 8l). These results demonstrate that sensitization of OSI-906 treatment to the apoptotic effect of LSD1 inhibition is due to the OSI-906 - dependent down-regulation of CFLAR expression.

## LSD1 inhibition synergizes with CR in other AMLs and breast cancer

To verify whether the synergism between CR and LSD1 inhibition can be exploited in other, more common neoplasms, first we explored the impact of CR and LSD1-inhibition in a PDX of an AML bearing an MLL-AF9 translocation, transplanted in NSG mice and CR-unresponsive. As shown in Fig. 6a, the addition of LSD1i resulted in significantly increased survival.

To extend to solid tumors the same therapeutic strategy, we explored the correlation between CFLAR dependency and *Lsd1(Kdm1a)* or *Oas1* expression in the DepMap dataset, which allows the intersection of quantitative estimates of gene dependencies (based on CRISPR–Cas9 essentiality-screens) with transcriptional data[60] across hundreds of cell lines[60]. We found that cancer cells with lower levels of *Lsd1* or higher levels of *Oas1* or *Oas3* were significantly more dependent on CFLAR (Fig. 6b and Supplementary Fig. 9a, b). Correlation with RNASEL expression was not explored because RNASEL levels are typically not affected transcriptionally by dsRNA[61]. Stratification of the DepMap analysis by originating anatomical location revealed a significant correlation within breast tumors (Supplementary Fig. 9c).

Thus, we tested the effect of CR on patient-derived xenografts (PDX) of triple-negative breast cancer patients available in our lab and characterized for their genomic profile by Whole Exome Sequencing[62] (Supplementary Data 2). CR slowed the growth of 2/3 PDX tested, with

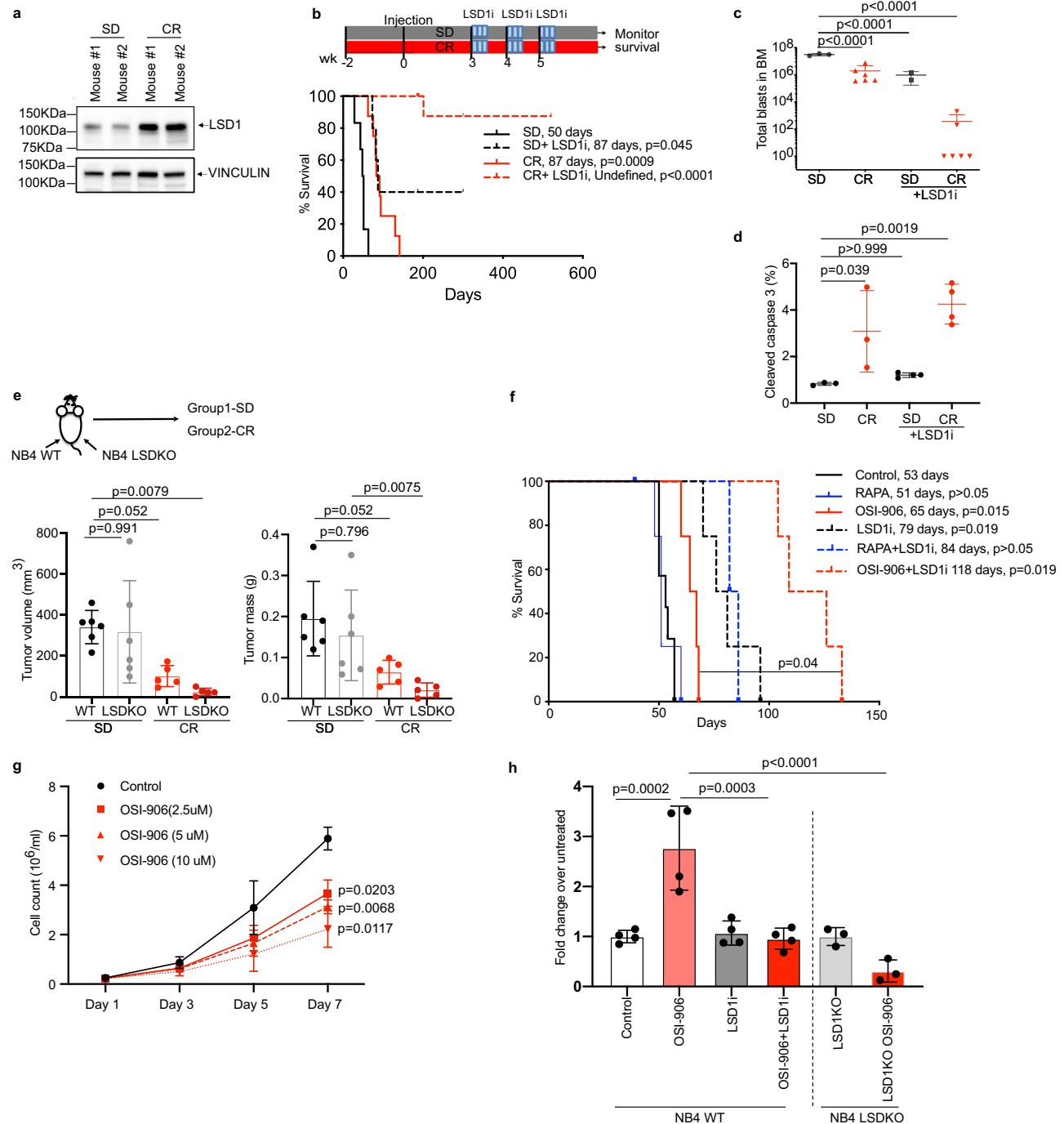

**Fig. 3 | LSD1 inhibition eradicates leukemia in CR-treated mice. a** LSD1 expression in CR and SD in the BM analyzed by WB using specific anti-LSD1 antibody (representative of 3 independent experiments). **b** Effect of LSD1i on survival of APL-bearing mice. Schematic representation (upper panel) and Kaplan−Meier survival curve (lower panel) after LSD1 inhibition under SD and CR diet. Median survival in days and *p* values, with SD as a common control (log-rank test, Bonferroni-adjusted) are indicated. SD+LSD1i (*n* = 5), CR (*n* = 8; vehicle), CR+LSD1i (*n* = 9), SD (*n* = 6; vehicle). Quantification of total numbers of APL-blast (in **c** by FACS; SD, *n* = 3; CR, *n* = 6; SD+LSD1i, *n* = 2; CR+LSD1i, *n* = 6) and frequency of apoptotic cells (in **d**; SD, *n* = 3; CR, *n* = 3; SD+LSD1i, *n* = 4; CR+LSD1i, *n* = 4; the SD and CR group is same in Supplementary Fig 1l. **e** Impact of CR on growth of NB4 cells. Upper panel: experimental scheme. NSG mice injected subcutaneously with $10^6$ WT (wild-type) (left flank) or LSD1 KO (knockout) (right flank) NB4 cells, fed SD (*n* = 6) or CR (*n* = 5), sacrificed after 20 days to measure tumor volume (lower left panel) and mass

(lower right panel). **f** Kaplan−Meier survival curves of APL recipient mice subjected to vehicle (40% PEG), rapamycin (RAPA) (4 mg/kg) or OSI-906 (20 mg/kg), alone or in combination with LSD1i (DDP37368) (45 mg/kg). The median survival in days and *P* values, with SD as a common control (log-rank test, Bonferroni-adjusted) are indicated. RAPA (*n* = 4), OSI-906 (*n* = 4), SD+LSD1i (*n* = 4), RAPA+ LSD1i (*n* = 3), OSI-906 + LSD1i (*n* = 4) with SD (*n* = 7; vehicle). **g** Effect of OSI-906 treatment on NB4-cell growth in vitro (*n* = 3 biological replicates). Data are expressed as mean ± SD, one way (for **c**−**e**, **h**) and two-way (**g**) ANOVA with post-hoc Tukey's multiple comparison. 'n' represents number of mice/groups for **b**−**f**, adjusted p values indicated. **h** Effect of OSI-906 treatment and LSD1 ablation on NB4 colony formation (two separate sets of experiments, *n* = 4 biological replicates for WT NB4, left panel, *n* = 3 biological replicates for LSD1 KO cells, right panel). For **h**, data are normalized on the average of the relative untreated control and expressed as mean ± SD. Source data are provided as a Source Data file.

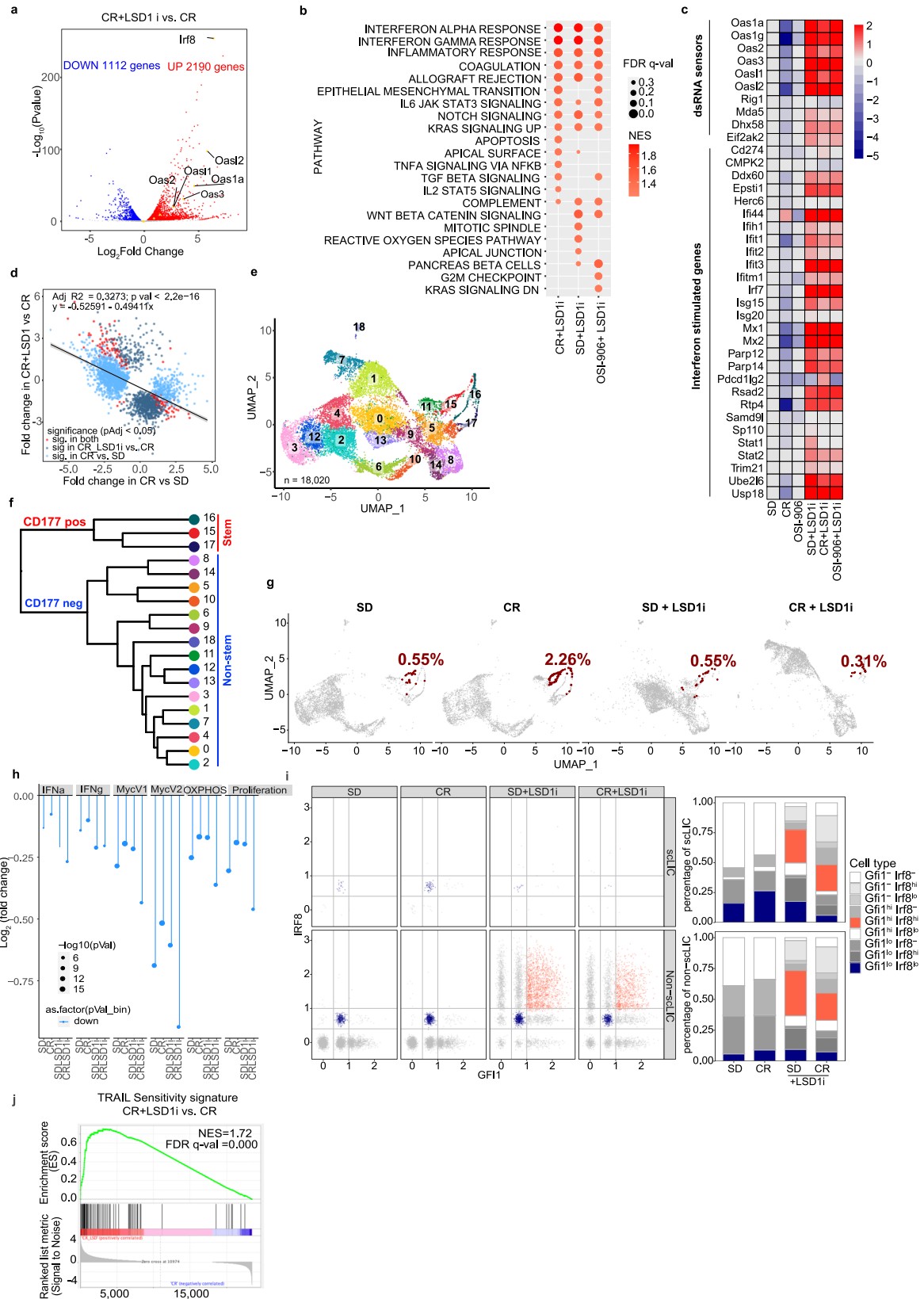

the nonresponsive PDX bearing a PIK3CA mutation, in agreement with previous findings[5] (Supplementary Fig. 9d–f). Co-treatment of the two CR-responsive PDX with LSD1i led to a further decrease in tumor size, as compared to CR alone (Fig. 6b, c). Thus, the synergy between CR and LSD1 inhibition is not specific to APL but can be observed in multiple solid and hematological neoplasms.

## Discussion

Our results reveal a yin-yang effect of CR on CR-responsive tumors: despite eliciting short-term tumor growth inhibition, CR favors survival of cells with stem-cell features that cause the delayed emergence of a more aggressive disease. The "persistent" phenotype is associated with and possibly caused by the LSD1-dependent downregulation of

**Fig. 4 | LSD1 inhibition de-represses endogenous-dsRNAs expression and reduces numbers of CD177 + APL cells with stem-like properties. a** Volcano plot showing differentially expressed genes in CR+LSD1i and CR APL blasts. Genes in red/blue are significantly up/down regulated (FDR < 0.05). Relevant ISGs are highlighted. **b** Top hallmark gene sets (from MSigDB) enriched in SD, CR and OSI-906 APLs treated with LSD1 inhibitor. The color of the dot indicates the normalized enrichment scores (NES) and size of the dot indicates the FDR q values. Only dots with an FDR < 0.25 are reported. **c** Heatmap of log2 fold-change regulation in dsRNA-sensing and ISGs genes relative to SD untreated APLs. The color scale indicates the log2 fold-change. **d** Linear regression analysis of fold change regulation in ERV expression showing LSD1-inhibition - dependent de-repression of dsRNA-inducing ERVs in CR. **e** UMAP visualization of scRNAseq data from of 18,020 cells grouped in 19 clusters. Cells are colored by clusters. **f** Hierarchical clustering highlighting the first bifurcation on Cd177 positivity: stem-like clusters 15-17 vs all other non-stem clusters. **g** UMAP visualization of cells split by treatment conditions (SD, CR, SD+LSD1i, and CR+LSD1i, as indicated). *Cd177 + Cd34*+ cells are highlighted in red and their frequency *per* condition is indicated. **h** Average log2 fold change (scLIC vs non-scLIC) in genes belonging to the indicated Hallmark gene sets in the scRNAseq data, by treatment condition. **i** Expression of Gfi1 and Irf8 within the scLIC (*Cd34 + /Cd177hi*) and non-scLIC (non-*Cd34 + /Cd177hi*) cells in the scRNAseq. Left panel shows expression levels (normalized counts), right panels show the corresponding fraction of cells (expressed as percentage of normal/gated). **j** GSEA plots showing enrichment of TRAIL sensitivity signature upon DDP treatment in CR and SD samples. NES, normalized enrichment score. Source data are provided as a Source Data file.

exhaustion- and death-inducing innate immune pathways (IFN, dsRNA sensing).

Specific points of pathway crosstalk appear as targetable vulnerabilities in cancer cells. Importantly, targeting each of these hubs individually is insufficient to cause disease eradication. CR and, in particular, insulin/IGF1 inhibition slows down the cell cycle and thus global tumor growth, but at the same time attenuates responsiveness to innate immune pathways that would induce cancer stem-cell clearance, favoring their persistence. Thus, upon CR, pro-survival and death-promoting stimuli are equally tuned down, resetting the system to a lower degree of activity with slower cell proliferation but preserved survival. However, if dsRNA sensing is activated cell-intrinsically through LSD1 inhibition, cancer cells in CR are unable to sufficiently upregulate anti-apoptotic factors like CFLAR, due to a global decrease in translation rate[58], and eventually die. Our data suggest that none of the main signals emanating from insulin/IGF1 receptor (PI3K, MEK, mTOR) is, by itself, sufficient to sensitize cells to LSD1 inhibition and dsRNA-mediated cell death. However, concerted inhibition of at least PI3K and MEK is required. Recent studies show evidence of synergism between mTOR and LSD1 inhibition in other systems[63,64], so further research is needed to clarify the specific dependence.

Our scRNAseq data suggest that in mouse APL, LICs are identified by the expression of CD177 and low-moderate levels of the opposing transcription factors Gfi1 and Irf8. Similarly, to what was observed in normal myeloid precursors[51], we do not observe Gfi1hi/Irf8hi cells in unperturbed conditions, but only shortly after LSD1 treatment, i.e., in conditions that lead to significant subsequent cell death, further suggesting that the double-high state is not compatible with cell survival.

Combining LSD1 inhibition with modulation of insulin/IGF1 signaling or other metabolic pathways may be a viable strategy to obtain disease eradication in patients.

Sensitivity to this strategy is likely determined by genetic features[5], as previously shown[5] although our data suggest that some genetic lesions that render tumors refractory to CR alone (e.g., MLL-AF9[23]) may still permit the response to combined CR+LSD1i. Agents targeting LSD1[44], Insulin/IGF1R[65] and/or PIK3CA + MEK[66,67] are already approved or in clinical development for numerous oncology indications. Testing the clinical utility of combining existing agents is feasible, particularly in neoplasms in which LSD1 inhibition has already shown signs of efficacy, like AMLs and neuroendocrine tumors including challenging contexts like SCLC. Measuring LSD1 upregulation upon dietary or pharmacological intervention may help identify putative responders.

## Methods
### Ethical statement
This research work complies with all relevant ethical regulations. All the animal experiments were approved by Italian Ministry of Health (Project no. 1072/15 and 833/18), conducted in accordance with the Italian law and under the control of institutional (European Institute of Oncology) local animal welfare (Cogentech OPBA) and ethical committee.

### Compounds
DDP37368 (molecule14d) and its enantiomers DDP38003 (molecule15) has been synthesized as previously described[68]. Details of other chemicals used in this study are Caspase-8 Inhibitor Z-IETD-FMK (R&D System, cat#FMK007), Caspase-3 Inhibitor Z-DEVD-FMK (R&D System, cat#FMK004), Pan Caspase Inhibitor Z-VAD-FMK (R&D System, cat#FMK001), m-TOR inhibitor rapamycin (LC Laboratories, R-5000 Rapamycin), IGF-1R/IR inhibitor OSI-906 (cat# S1091, Selleckchem.com), PI3K inhibitor Ly294002 (cat#9901, Cell Signaling Technology), ERK inhibitor Trametinib (GSK-1120212, cat#A-1258, Active Biochem), cycloheximide (Sigma, cat#C7698), actinomycin D((CAS 50-76-0); Santa Cruz Biotechnology, Inc., cat# sc200906) and TRAIL (ALX-201-073-C02, 3v Chimica Srl).

### Cell lines
NB4 cells, derived from a female APL patient (RRID: CVCL_0005) were a kind gift from Dr M. Lanotte (INSERM, Paris, France). Generation of NB4 LSD1 KO cells by CRISPR-Cas9 was previously described[69]. NB4 and NB4 LSD1 KO cells were maintained in RPMI 1640 medium (Gibco BRL, Paisley, UK) supplemented with 10% fetal bovine serum (FBS) (Euroclone, UK), 2 mM L-glutamine, 100 U/ml penicillin and 100 mg/ml streptomycin (Euroclone, UK) at 37 °C under 10% $CO_2$. HEK-293T cell line was acquired from ATCC (RRID: CVCL_0063) and are available in the IEO cell line repository. Cell lines were routinely tested for the mycoplasma infection. HEK-293 T cells were cultured in DMEM supplemented with 10% FBS, 2 mM glutamine, 100 U/mL penicillin and 100 μg/mL streptomycin.

Cell lines are periodically tested for authentication using the ProMega geneprint 10 PCR-based kit (cat n B9510). Cells are never kept in culture for longer than a month or 10 passages, whichever is shortest. Cell lines are periodically tested and for Mycoplasma contamination via the Uphof/Drexler protocol at every new restock, on average every 3 months.

### In vitro maintenance and culturing of mouse NPM1c+ blasts
Spleen-derived NPM1c+ blasts were isolated from fully engrafted mice spleens through smashing and filtering process (70 μm filter). NPM1c+ blasts were frozen in cryovials at the concentration of $20 × 10^6$/ml in FBS (Euroclone, UK) containing 10% DMSO. Thawed cells were maintained between $0.5 × 10^6$ and $2 × 10^6$/ml in RPMI medium supplemented with 12.5% calf serum, 12.5% horse serum, 2 mM glutamine, 50 mM beta-merkaptoethanol, 1 mM hydrocortisone, 10 ng/ml interleukin 10 ng/ml interleukin and 50 ng/ml stem cell factor. Cells were grown in humified incubator with normal oxygen and 5% CO2.

### NB4 colony formation
NB4 or NB4 LSD1KO cells were plated at a density of 250,000 cells/ml in the presence of 0.1% DMSO or 5 μM OSI-906 ± LSD1i 2.5 μM, and

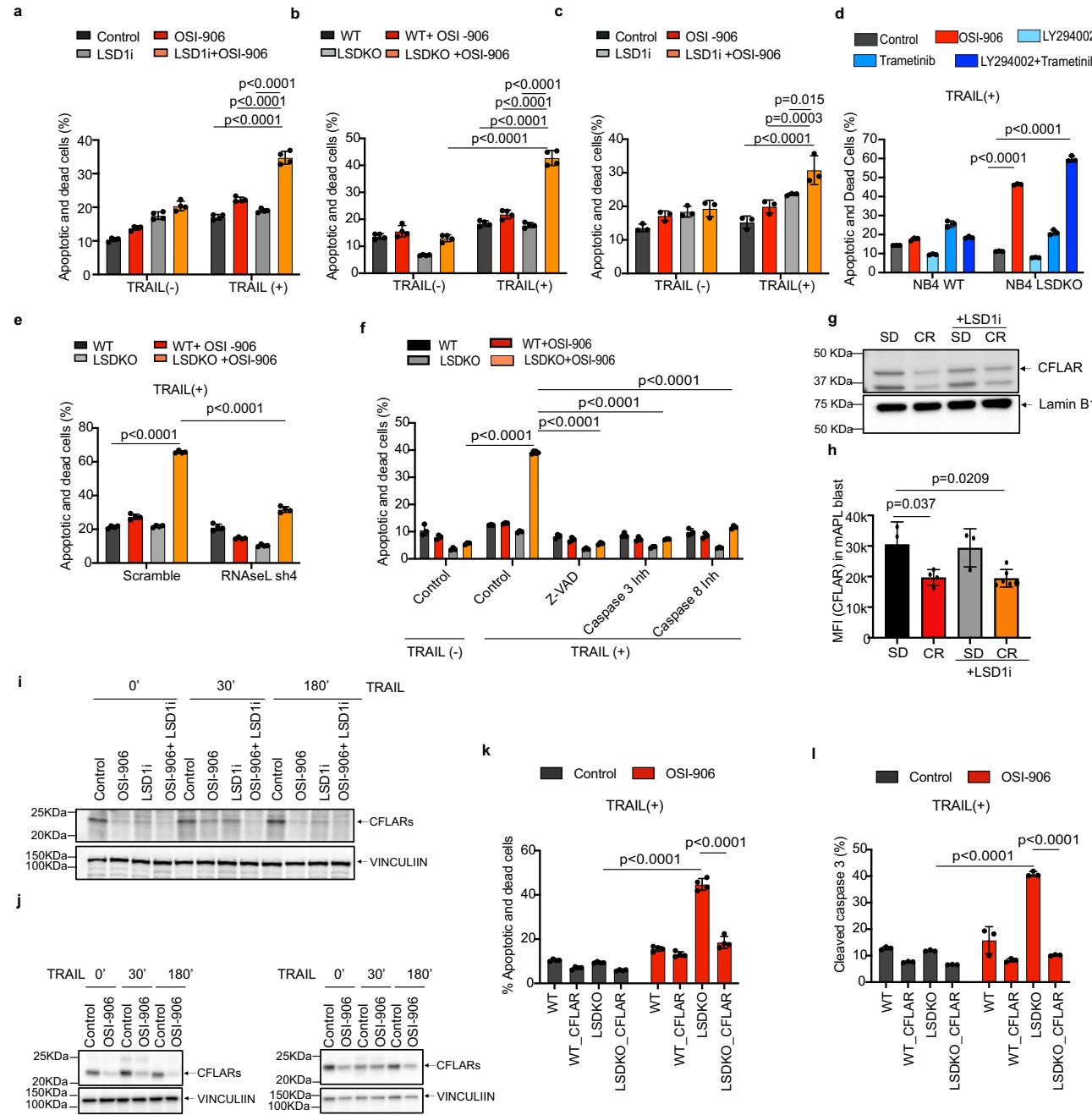

**Fig. 5 | Combined insulin/IGF1 and LSD1 inhibition sensitizes cells to RNASEL and CFLAR-dependent TRAIL -induced cell death. a–f** Percentage of apoptotic and dead cells in different experimental conditions: pharmacologically inhibited (**a**) or genetically depleted (**b**) LSD1 in NB4 cells, ±OSI-906 (5 µM) for 48 h followed by addition of TRAIL (100 ng/ml) for 20 h, (*n* = 4 biological replicates, representative of 4 independent experiments); **c** Percentage of apoptotic/dead cells (by Annexin V and PI staining) of LSD1i- (2.5 µM; 24 h) and/or OSI-906 (10 µM; 24 h) treated or untreated NPM1c+ blasts, ±TRAIL (100 ng/ml) for 20 h (*n* = 3 biological replicates, representative of 3 independent experiments). **d** LSD1 WT and KO NB4 cells treated with TRAIL and OSI-906 (5 µM), PI3K inhibitor Ly294002 (10 µM) and ERK inhibitor Trametinib (4 nM), alone or in combination, (*n* = 3 biological replicates, representative of 3 independent experiments); **e** LSD1 WT or KO cells, with shRNASEL or control, ±OSI-906 (*n* = 4 biological replicates, representative of 3 independent experiments); **f** LSD1 WT or KO cells, ±TRAIL or OSI-906 and different pan- or specific caspase inhibitors (*n* = 3 biological replicates, representative of 4 independent experiments). **g, h** CFLAR expression on APL blasts from mice exposed to SD or CR diet ±LSD1i, by WB (**g**) or FACS (SD, *n* = 3; CR, *n* = 4; SD+LSD1i = 3; CR +LSD1i, *n* = 6) ("*n*" number of mice). (**h**). MFI: Mean Fluorescence Intensity. **i, j** CFLAR expression in NB4 cells ±LSD1i ±OSI-906 (**i**) and NB4 LSD1 WT/KO cells ±OSI-906 (**j**), ±TRAIL for indicated time by WB. Note that the CFLAR migration appears delayed in NB4 LSDKO cells on the gradient gel, suggesting modification upon LSD1 ablation; **k–l** Effect of CFLAR overexpression on TRAIL-induced death of NB4 LSD1 WT or KO cells ±OSI-906, by annexin/PI staining (**k**, *n* = 4 biological replicates, representative of 3 independent experiments) and cleaved caspase 3 staining (**l**, *n* = 3 biological replicates, representative of 2 independent experiments). Data are expressed as mean ±SD, two-way ANOVA with post-hoc Tukey's multiple comparison. Western blots are representative of 3 independent experiments. Source data are provided as a Source Data file.

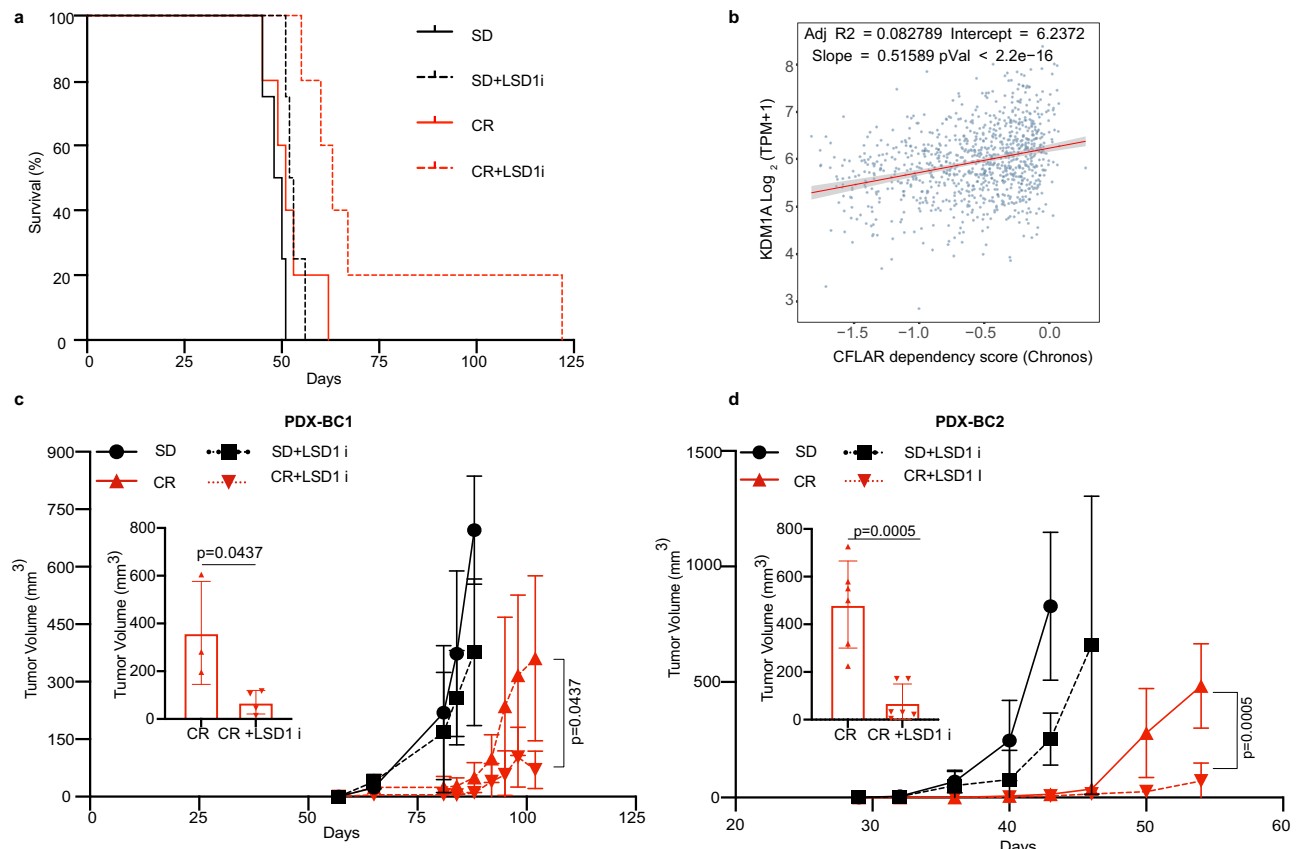

**Fig. 6 | LSD1 inhibition synergizes with CR in other AMLs and breast cancer.**
**a** Kaplan–Meier survival curve of leukemia-bearing mice fed SD or CR and treated or not with LSD1i: AML-IEO20 PDX with the MLL-AF9 translocation (SD, $n = 4$; CR, $n = 5$; SD+LSD1i, $n = 4$; CR+LSD1i, $n = 5$) ($p = 0.02$; CR +LSD1i vs. CR). "n" represents the number mice per group. Statistical analysis was performed using log-rank (Mantel–Cox) test. **b** DEPMAP analysis of CFLAR dependency vs LSD1 (KDM1A). Linear regression parameters are overlayed. Impact of CR and LSD1 inhibition on tumor growth of PDX-BC1 (**c**) and PDX-BC2 (**d**) over time. Inserts show tumor growth at the end of the observation period (102 days for PDX-BC1 and 54 days for PDX-BC2 for CR and CR+LSD1i group as indicated. For PDX-BC1 (SD, $n = 5$; CR, $n = 3$; SD+LSD1i, $n = 6$; CR+LSD1i, $n = 4$) and for PDX-BC2, (SD, $n = 8$; CR, $n = 6$; SD+LSD1i, $n = 8$; CR+LSD1i, $n = 6$) where "$n$" is the number mice per group. Mice with non-palpable tumor during the entire experiment were removed from the analysis. Data are shown as mean ± SD. Statistical analysis was performed using two tailed t-test between CR and CR+LSD1i group. Source data are provided as a Source Data file.

counted every 2 days. After 7 days, they were resuspended in cytokine-free methylcellulose H4230 (Stem Cell Technologies) at 2000 cells/ml in triplicate wells, in the presence of the same doses of OSI-906 and LSD1i. Colonies were scored after 1 week upon staining with MTT (3-(4,5-dimethylthiazol-2-yl)−2,5-diphenyltetrazolium bromide) (M5655, Sigma).

**Gene knockdown by shRNA**
In order to knock-down *RNaseL*, three short harpin RNA (shRNA) sequences were used. Interfering sequences were cloned by AgeI-EcoRI double digestion into the modified pLKO.1 vector (Addgene plasmid # 8453) that was generated by replacing the cDNA sequence encoding for the puromycin selection marker with the cDNA encoding for the selection marker Enhanced Green Fluorescence Protein (EGFP). The interfering sequences used are the follows:

RNaseLsh2 (CCGGGACAATCACTTGCTGATTAAACTCGAGTTTA ATCAGCAAGTGATTGTCTTTTTTG), RNaseLsh3 (CCGGCTGAAGGA TCTCCACAGAATACTCGAGTATTCTGTGGAGATCCTTCAGTTTTTTG), RNaseLsh4, (CCGGGCTAAAGTTCATCCGGAATTTCTCGAGAAATTCCG GATGAACTTTAGCTTTTTG). Lentivirus carrying shRNAs or scrambled was produced by co-transfecting HEK-293T cells with vesicular sto-matitis virus glycoprotein and dR8.2 and by harvesting viral super-natant after 48 h by passing through a 0.45 μm filter. NB4 WT cells and NB4LSDKO cells expressing shRNAs were generated by lentiviral transduction using 8 μg/ml polybrene. After a recovery period of 2–5 days, cells were sorted by fluorescence-activated cell sorting (FACS) using the GFP selection marker and maintained in the culture for further experiments.

**Ectopic expression of CFLAR**
CFLAR short (clone RC225270) cDNAs constructs were purchased from Origene. Constructs were PCR-amplified from original vectors using the following primers: forward AAAGCTAGCGCCGCCGC-GATCGCCATGTCT, reverse AAATTCGAACTTATCGTCGTCATCCTT GTA. PCR products were then cloned into the lentiviral vector pCDH-MCS-T2A-copGFP-MSCV for ectopic expression, using NheI and BstBI restriction enzymes. The virus particle was produced and used to transduce NB4 WT and NB4LSDKO, followed by selected sorting of GFP+ transduced cells.

**ELISA for IGF1**
The ELISA assay was performed with a mouse IGF1 DuoSet Elisa development kit (R&D system, cat# DY791) according to the manu-facturer's instructions.

**In vitro apoptosis measurement**
Cell apoptosis was quantified by flow cytometry using eBioscience™ Annexin V-FITC/PI (ThermoFisher Scientific, cat# 88-8005-74) or Annexin V-APC/PI (for GFP- expressing cells; ThermoFisher Scientific, cat# 88-8007-74) Apoptosis Detection Kits following manufacturer's

instructions. For, each of the cell lines, after the first splitting the cells were divided into three or four seeding flask that were maintained for a week separately before the start of the experiment. From each seeded flask, cells were pre-treated with OSI-906(5 μM) or LSD1i (2.5 μM) or both for 48 h. Activation of cleaved caspase 3 was quantified after 6 h and apoptosis was quantified after 20 h after the addition of Trail (100 ng/ml) in the culture medium. In order to determine the involvement of caspase-3 and caspase-8 dependent apoptotic pathway, caspase-8 inhibitor Z-IETD-FMK (20 μM), caspase-3 inhibitor Z-DEVD-FMK (20 μM), pan caspase inhibitor Z-VAD-FMK (20 μM) was added with TRAIL. For the involvement of PI3K and MAPK pathways, after 48 h of pretreatment with Trametinib (4 nM) or Ly294002(10 μM) or both, apoptosis was quantified after 20 h after the addition of Trail (100 ng/ml). For the inhibition of translation or transcription LSD1 WT/KO cells were pre-treated with non-lethal doses of actinomycin D (10 ng/ml) or cycloheximide (1 μg/ml) followed by treatment with TRAIL (100 ng/ml) for 20 h. NPM1c+ blasts were pre-treated with OSI-906 (10 μM) or LSD1i (2.5 μM) or both for 24 h. Apoptosis was quantified after 20 h after the addition of TRAIL (100 ng/ml) in the culture medium.

## Mice and treatments

7–8-week C57BL6J (Ly5.1 and Ly5.2), NOD.Cg-Prkdc[scid]Il2rg[tm1Wjl]/SzJ (NSG) and RAG−/− mice were purchased from Charles River Laboratories Italia and maintained in our animal facility (European Institute of Oncology Cogentech Facility) under specific pathogen-free conditions. Experiments were conducted upon approval of the local Animal Welfare Committee and Ministerial Project no. 1072/15 and 833/18. Specific sexes were used for each experiment, to match the genetic sex of the transplanted tumor. In particular, all mice used as APL recipients were male, to match the sex of the APL transplantable clone. NSG mice used for TNBC PDX were all female. All mice were fed (VRF1 (P); Special Diet Services; #801900) either ad libitum (AL) or with CR regimen (70% of the normal food intake). LSD1 inhibitor (DDP37368) was administered by oral gavage at 45 mg/kg (in 40% PEG,400) of body mass three days/week for three weeks. LSD1 inhibitor (DDP38003) was administered oral gavage at 17 mg/kg (in 40% PEG, 400) of body mass two days/week for three weeks. Rapamycin (in DMSO) was injected intraperitoneally with 4 mg/kg of body mass five days/week for 4 weeks. 20 mg/kg of OSI-906 diluted in DMSO was administered every other day by oral gavage for 4 weeks. Treatment with rapamycin and OSI-906 was started 2 weeks after the transplant and treatment with LSD1 inhibitor was started 3 weeks after the transplant. In case of AML-IEO20 (PDX) treatment with LSD1 inhibitor was started after a week of transplant.

## In vivo AML transplantation model

The transgenic mouse model PML-RARA APLs in the C57BL/6-Ly5.2 background was described previously[70]. These mice develop acute myeloid leukemia after long latency. The leukemia that develop in these mice can be transplanted serially and retain its similarities with human APLs in the transplanted mice. The in vivo APL transplantation model has been used routinely to do drug sensitivity assay and target identification[71]. For the development of APL in mouse, we used a passage 3 APL developed in a male mouse and serially expanded through intravenous injection and spleen harvesting, routinely confirmed to be >95% infiltrated prior to subsequent passage. For experiments, 1-2 × 10⁶ APL cells in 1X PBS were intravenously injected into congenic 8 wk old (C57BL/6-Ly5.1) recipients. Similarly, mouse models of AMLs were obtained by transplanting NPM1c+ and NPM FLT3-ITD expressing blasts obtained from the transgenic mice generated in our group[22]. MLL-AF9 (MA9) leukemia was generously provided by Dr. Chi Wai So[72] and transplanted in C57BL/6-Ly5.2 recipients. In brief, 0.2–2 ×10⁶ leukemic blasts in 1X PBS were intravenously injected into congenic 8–10 wk old (C57BL/6-Ly5.1) recipients which were

under CR for 2 weeks or mice on an ad libitum diet. In some cases, mice were randomly divided into CR and SD group, 4 days or 4 weeks after the transplantation. Mice were maintained on CR throughout the experiment. The latency in leukemia development was monitored and mice were sacrificed when moribund, following the animal facility guidelines. The C57BL/6 congenic differs in pan leukocytes marker Ptprc, known as CD45 can be used to differentiate host cells with the injected APL cells. The progression of APL in the mouse was monitored each week by analyzing the presence of CD45.2 APL cells in peripheral blood.

## Isolation of cells

In order to determine the progression of leukemia, blood was collected weekly from APL transplanted mice from the tail vein in EDTA (final concentration 3 μg/μl) and was lysed with RBC lysis buffer (150 mM NH4Cl, 15 mm KHCO3, 126 μM EDTA). The cells were washed twice and resuspended in in 2% fetal bovine serum in 1X PBS (FACS Buffer). Bone marrow cells were obtained by flushing FACS buffer through mouse hind limbs bones. RBC was lysed using RBC buffer and remaining BMNC cells were washed and filtered through a 70-micron filter and resuspended in FACS buffer for flow cytometric analysis or in 1XPBS for transplantation.

The spleen was excised into small pieces and pressed through the strainer using the plunger end of a syringe. Splenocytes were obtained and suspended in FACS buffer after removal of RBC using RBC lysis buffer, washed, and filtered through the strainer.

## Cell surface and intracellular staining

Isolated cells from bone were first blocked with 10% BSA to avoid non-specific binding and then stained with a fluorochrome-conjugated monoclonal antibody directed against CD 45 Ly5.2(from eBioscience clone 104) to distinguish leukemic blast and antibodies against CD34, c-Kit, Gr1, FcγRIII/II and CD177. For the intracellular staining BMNC were fixed in Cytofix/Cytoperm buffer, permeabilized in BD cytoperm plus buffer, followed by refixation in Cytofix/Cytoperm buffer. Fixed cells were stained with intracellular antibodies in perm wash buffer 1 hr at room temperature/ overnight at 4 degrees. For CFLAR and LSD1 detection, after primary antibody incubation and washes, cells were incubated with fluorochrome-conjugated secondary antibody. For the cell cycle analysis, cells were stained with DAPI and analyzed by flow cytometry.

## O-propargyl-puromycin labeling

NB4 cells were pre-treated with OSI-906 (2.5 μM and 5 μM) for 24 h. Next day, the cells were plated in 96 well plate at the density of 500,000 cells/ml. Click-iT® OPP was added at a concentration of 20 μM while maintaining the concentration of OSI-906. Cycloheximide was added at a concentration of 50 mg/ml as a control. NB4 cells were incubated for 20 min in the incubator at 37 °C under 10% CO₂. Cells were washed once with PBS, fixed in 4% paraformaldehyde and permeabilized using 0.5% Triton X-100 in PBS. According to the manufacturer's direction click reaction was carried out with Click-iT Plus OPP Alexa Fluor 647 Protein Synthesis Assay kit (Thermo Fisher; # C10458).

## Flow cytometry

FACS CantoII and FACS Celesta flow cytometer (BD Biosciences, Oxford, UK) were used for multi-parametric and cell-cycle flow cytometry data acquisition.

Cell sorting experiments were performed using a FACSAria Fusion cell sorter (BD Biosciences, Oxford, UK). The antibodies used for flow cytometry are CD45.2 monoclonal antibody (104), FITC (eBioscience™; #11-0454-82, dilution 1:100); CD45.2 monoclonal antibody (104), PE (eBioscience™; #12-0454-82, dilution 1:100); CD45.2 monoclonal antibody (104), APC (eBioscience™; #17-0454-82, dilution 1:100); CD45.2

monoclonal antibody (104), APC-Cy7 (BD Pharmingen™; #560694, dilution 1:100); CD45.1 monoclonal antibody (A20), PE-Cyanine7 (eBioscience™; #25-0453-82, dilution 1:100); CD45.1 monoclonal antibody (A20), PE (eBioscience; # 12-0453-82, dilution 1:100); alexa fluor® 488 mouse anti-Ki-67 (B56)(BD Pharmingen™; # 558616, dilution 1:25); alexa fluor® 647 Mouse anti-Ki-67(B56) (BD Pharmingen™; #561126, dilution 1:25); PE rabbit anti- active caspase-3 (C92-605) (BD Pharmingen™; # 550821; dilution 1:10); CFLAR/FLIP (D5J1E) rabbit mAb (Cell signaling; #56343, dilution 1:100), CD177 antibody (R&D Systems; #MAB8186, dilution 1:150), CD34 monoclonal antibody (RAM34) FITC (eBioscience™, # 11-0341-82, dilution 1:100), FcγRIII/II (CD16/CD32) monoclonal antibody (93), eFluor™ 450 (eBioscience™, # 48-0161-82, dilution 1:100), GR1 (Ly-6G/Ly-6C) monoclonal antibody (RB6-8C5), PE, (eBioscience™, # 12-5931-82, dilution 1:100), CD117 (c-Kit) monoclonal antibody (ACK2), APC-eFluor™ 780, (eBioscience™, # 47-1172-82, dilution 1:100), Sca-1 monoclonal antibody (D7), PerCP-Cyanine5.5 (eBioscience™, # 45-5981-82, dilution 1:100), Flk2 (Flt3/CD135) monoclonal antibody (A2F10), APC (eBioscience™, # 17-1351-82, dilution 1:100); CD11b monoclonal antibody (M1/70), FITC, (eBioscience™, #11-0112-82) and anti-KDM1/LSD1 poly clonal antibody (abcam, # ab17721, dilution 1:100). In case of unconjugated primary antibody, additional incubation with fluorochrome conjugated secondary antibody was done after the washes. Secondary antibody used is alexa fluor 647 affinipure donkey anti-rabbit IgG (polyclonal, Jackson ImmunoResearch Laboratories, #715-605-152). Analyses were carried out using FlowJo v 10.

For cell sorting prior to RNA or scRNA sequencing, samples were stained with DAPI (for viability), CD45.2 and CD45.1 and DAPI-negative/CD45.2+/CD45.1− cells were sorted. Purity was checked by re-analyzing the sorted population and was routinely >98%. Cells were gated for live cells using the DAPI channel (DAPI-neg) and FSC/SSC, doublets were discriminated by FSC A vs FSC H, CD45.2 and CD45.1 gates were set based on pure spleen cell preparations from non-transplanted CD45.2 or CD45.1 mice.

Bone-marrow mononuclear cells (BMNCs) from healthy (i.e., non-APL-transplanted) mice that were subjected to CR for 4 weeks ±LSD1i for 3 days, were stained with appropriate antibodies and analyzed by FACS to identify populations of HSCs (LT-HSCs: long-term reconstituting HSCs, Lin-/c-Kit+/Sca-1+/Flk2−/CD34−) and progenitors (MPPs: multipotent progenitors, Lin−/c-Kit+/Sca-1+/Flk2+/CD34+), myeloblasts (Gr1+/CD11b-), myeloid derived suppressor cells (GR1+/CD11b+), myelocytes (Gr1-/CD11b+), and erythroblast (Ter119+). Lineage negative (Lin−) cocktail consisted of anti-mouse antibody against CD11b (eBioscience™, M1/70; # 25-0112-82), Gr1 (eBioscience™, RB6-8C5; # 25-5931-82), CD3 (eBioscience™, 145-2C11; # 25-0031-82), Ter119 (eBioscience™, Ter119; # 25-5921-82) and B220 (eBioscience™, RA3-6B2; # 25-0452-82) in PE-Cy7.

## Limiting dilution transplantation assay

For the limiting dilution transplantation assay, APL blasts (Ly5.2+) or NPM1c+ blast (Ly5.2) was sorted by the expression of CD45.2 antigen from freshly isolated BMNC of leukemic mice (Ly5.1+). Sorted cells were intravenously injected (from $2 \times 10^5$ to 200 cells per mice for APL blast and $5 \times 10^4$ to 50 cells per mice for NPM1c+ blast) into C57 BL/6J recipient mice, with $2 \times 10^5$ splenocytes as a carrier. Leukemia development and survival were determined and LIC frequency was measured using ELDA software[27] (https://bioinf.wehi.edu.au/software/elda/).

## Breast PDX in vivo study

Breast PDXs used in this study were generated and maintained, as previously described[62]. PDXs (obtained from passages in the NSG mice) were thawed, and $5 \times 10^5$ cells were orthotopically injected into the fourth mammary gland NSG mice. A week after the injection mice were randomly divided into SD and CR groups. Mice were monitored

for the tumor development and tumors were measured with calipers. Tumor volume was calculated using this formula: $V = 1/2$ (length × width$^2$). A maximum tumour size of diameter 1.5 cm was permitted by the ethics committee. The maximum tumor size was not exceeded. Treatment with drug was started when tumors became palpable in majority of the mice. LSD1i (DDP38003) was administered orally at the dose of 17 mg/kg × 2 days per week for a total of 3 weeks in a vehicle composed of 40% polyethylene glycol, molecular weight 400 in PBS.

## Immunohistochemitry and Hematoxylin/Eosin staining

Blood smears were stained with May-Grünwald-Giemsa (Sigma Aldrich) according to the Sigma-Aldric Giemsa stain protocol (Procedure No. GS-10). Femor was first fixed overnight in 4% formaldehyde, followed by decalcification using osteodoc (Bio-optica, Cat# 05-M03005) following the manufacturer's instructions. Spleen was fixed overnight in 4% formaldehyde. Fixed tissues were embedded in the parafilm with LogosJ Processor (Milestone). Sections of 3 micron were cut and left for an overnight incubation at 37 °C before staining. IHC staining for Ki67-SP6 and cleaved caspase3 antibodies was performed using Bond III IHC auto-stainer for full Automated Immunohistochemistry (Leica biosystems). For both antibodies antigen was unmasked with Tris-EDTA pH 9 (Bond Epitop Retrival Solution 2 Leica AR9640) followed by peroxidase blocking using Bond Polymer Refine Detection Kit (DC9800). Tissues were washed and then incubated with primary antibodies diluted in Bond Primary Antibody Diluent AR9352 Leica. Subsequently, tissues were incubated with polymer for 10 min and developed DAB Chromogen following by Hematoxylin as counterstain. Pictures of stained sections were acquired with the HistoFluo microscope (Leica DM6). Primary antibodies used: anti-Ki67monoclonal antibody (SP6) (Invitrogen, #MA5-14520, 1:200), anti-cleaved caspase3, monoclonal antibody (Asp175) (Cell Signaling, #9661; 1:200).

## Western blotting

Cells were lysed in RIPA Buffer (50 mM Tris-HCl pH 8.0, 150 mM NaCl, 0.1% SDS, 1% NP-40 and 0.5% deoxycholic acid) + Protease Inhibitor Cocktail (Roche, 11836170001) + Phosphatase Inhibitor Cocktail 2 (Sigma, P5726), quantified using Pierce BCA Protein Assay (Thermo Scientific, 23225) and then mixed with 5x Laemmli SDS reducing buffer before incubation at 95 °C for 5 min. Both homemade and precasted (4–20% and 4–15% Criterion™ TGX Stain-Free™ Protein Gel; Bio-Red) gradient gel was used. Samples were separated on SDS-PAGE and then subsequently blotted onto PVDF membranes following conventional protocols. In brief, the membrane was blocked in 5% BSA in TBS-T (0.1% Tween-20 in 1x TBS) for 1 h followed by overnight incubation with primary antibody at 4 °C and then subsequently incubated with the secondary antibody at RT for 1 h. The membrane was then washed thrice with TBST and incubated with Clarity Western ECL Substrate (Bio-Rad, 170-5060) followed by imaging with the ChemiDoc XRS+ Gel Imaging System (Bio-Rad). Antibodies used are polyclonal caspase-3 antibody (Cell signaling, #9662, dilution 1:1000), Vinculin (hVin-1), mouse mAb (Sigma-Aldrich, #V9131, dilution 1:5000), Actin, mouse mAb (AC-40) (Sigma-Aldrich, #A4700, dilution 1:500), OAS1 (D1W3A) Rabbit mAb (Cell signaling, #14498, dilution 1:1000), MDA-5 (D74E4) Rabbit mAb (Cell signaling, # 5321, dilution 1:1000), RIG-I (D14G6) Rabbit mAb (Cell signaling, #3743, dilution 1:1000), RNASE L (D4B4J) Rabbit mAb (Cell signaling, #27281, dilution 1:1000), CFLAR/FLIP (D5J1E) Rabbit mAb (Cell signaling, #56343, dilution 1:1000) and anti-KDM1/LSD1 poly clonal antibody (abcam, # ab17721, dilution 1:1000). Secondary antibodies used are horseradish peroxidase (HRP)-conjugated anti-mouse IgG (Cell signaling, #7076, dilution 1:3000) and anti-rabbit IgG conjugated to HRP (Cell Signaling, # 7074, dilution 1:3000) for chemiluminescent detection.

## Whole exome sequencing

Genomic DNA was extracted using DNeasy Blood & Tissue Kits (QIA-GEN, Cat. No./ID: 69556) from the leukemic blast harvested from the bone marrow of moribund mice following the manufacturer's instructions. Exome capture was performed using SureSelectXT Mouse All Exon kit (Agilent Technologies) following the manufacturer's specifications. Whole-exome sequencing was performed with the Illumina NextSeq500, and NovaSeq6000 platform with 100 bp and 101 bp paired-end reads respectively. Alignment to the reference genomes (mm10) was performed using Burrows Wheeler Aligner-MEM v0.7.8[73] with default parameters. Aligned SAM files were converted to BAM files and sorted by coordinate with Samtools v0.1.18[74].

WES data have been then processed according to GATK v2.8-1[75] best practices through duplicate marking and base quality recalibration. In particular, the MarkDuplicate function of Picard (http://broadinstitute.github.io/picard) was applied to remove duplicated reads from each BAM file. We obtained a mean coverage of ~90X (ranging from 69X to 119X) and coverage greater than 10X for ~95% of the targeted exonic regions. We identified single nucleotide variants (SNVs) in our samples using MuTect.v1.1.4[76]. We applied to the list of variants detected the following filters: *i)* a minimum coverage of 10 reads for both control and diet-conditioned samples; *ii)* a minimum read depth of 3 for the alternate variant allele in at least one of the matched samples; *iii)* no reads supporting the alternative allele in control; *iv)* a variant allele in at least 10% of all reads covering the position; *v)* no common mutations with other samples of the same diet condition. Variant effect prediction was carried out using VEP release 109 (https://www.ensembl.org/info/docs/tools/vep/index.html), after liftover of genomic coordinates to GRCm39.

The identified variants were functionally annotated using Annovar v2015[77]. We excluded from further analysis variants in non-coding regions. The source list for Cancer Genes is ICGC/TCGA Pan-Cancer Analysis of Whole Genomes Consortium[78].

## RNAseq sample preparation and analysis

Bone marrow cells were stained with a fluorochrome-conjugated monoclonal antibody directed against CD 45 Ly5.2 and APL blasts were selectively sorted for the presence of CD45.2 by BD FACSAria. APL blasts were directly sorted in the extraction buffer, and RNA was extracted using Arcturus PicoPure RNA Isolation Kit (Thermo Fisher Scientific, KIT0204) according to the manufacturer's protocol. The RNA was used to construct cDNA libraries using the TruSeq RNA library kit (Illumina, San Diego, CA, USA) and libraries were sequenced in multiplex at 50 bp pair-end reads on an Illumina HiSeq2000 (average sequence depth ~36million reads/sample). For the stranded RNA, libraries were prepared using TruSeq stranded total RNA kit (Illumina, San Diego, CA, USA). Libraries were sequenced on an Illumina HiSeq2000 at 50 bp PE mode (average sequence depth ~36 million reads/sample). Sequences were aligned with Tophat2[79] against the *Mus musculus* reference genome (UCSC, release mm10). At the gene level, expression counts were estimated using HTSeq (v0.5.3p9)[80], summarized across all exons as annotated in the mouse transcriptome (downloaded from UCSC, mm10, archive-2013-03-06-15-06-02) with default options. Normalization and identification of differentially expressed genes in two biological replicates of standard diet (SD) control condition and in caloric restriction (CR) or in combination with other treatments were carried out using EdgeR R-package[81].

Functional annotation analysis was performed using GSEA v3.0. Transcripts were ranked by the Signal-to-noise metrics and using the following settings: a number of permutations = 1000, permutation type = gene set, enrichment statistic = weighted, gene list sorting mode=real, gene list ordering mode=descending, max gene set size = 500, min gene set size = 15. The curated gene sets collection (h.all.v7.0.symbols.gmt) was interrogated. Plots and heatmaps on the RNA-seq analysis part were drawn using the ggplot2 R package. For the heat-map of genes involved in ISG and ds-RNA signaling pathways the fold change obtained for each group with respect to SD was transformed into log2fold change for the visualization.

## scRNAseq sample preparation and analysis

For scRNA-seq libraries, we sorted APL blast from the bone marrow in 1xPBS containing 0.04% BSA. We submitted ~5000 cells for the Gel Bead-In-EMulsions (GEMs) using 10X Genomics Chromium machine. The libraries were prepared following the Single Cell 3'Reagent Kits (v2) user guide (manual part no. CG00052 Rev C). The kits used for the library preparation included Chromium Single Cell 3'Reagent Kits (v2): Single Cell 3'Library & Gel Bead Kit v2 (PN-120237), Single Cell 3'Chip Kit v2 (PN-120236) and i7 Multiplex Kit (PN-120262) (10x Genomics). Libraries were sequenced on NovaSeqTM 6000 Sequencing System (Illumina®) with an asymmetric paired-end strategy (28 and 91 bp read length for R1 and R2 mate respectively) with a coverage of about 50,000 reads/cell. After demultiplexing FASTQ files were converted to gene-cell count matrices using a Singularity pipeline employing the Cell Ranger v.4.0 software[82]. As reference, the *Mus musculus* reference genome mm10 (GENCODE vM23/Ensembl98) was used. We then merged the transcript count matrices from all four conditions (SD, CR, SD+LSD1i, CR+LSD1i) using Seurat v4[83]. Quality metrics on cells and genes were calculated separately for each replica for each condition. We identified putative doublets by mean of scDblFinder package and we removed them. Then, genes which were expressed in fewer than 5 cells were removed from the analysis. Similarly, cells with number of UMIs and number of genes detected per cell below median minus 5 times the MAD or higher median plus 5 times the MAD were discarded.

Next, we log-normalized the raw transcript counts matrix using SCTransform() function; then, principal component analysis (PCA) was performed on the resulting normalized data set. The Uniform Manifold Approximation and Projection (UMAP) for non-linear dimensional reduction purposes was employed on the first 15 principal components using the RunUMAP() function of the Seurat package.

In order to partitioning cellular distance matrix into clusters, we determined the k-nearest neighbor graph with FindNeighbors() function and then performed the clustering with the FindClusters() using default resolution parameter. Similarity between clusters was calculated and represented in a tree with the aid of the BuildClusterTree() funtion; the tree is estimated based on a distance matrix constructed in the PCA space. Cell-cycle annotation was performed using the CellCycleScoring() function by mean of S- and G2M-specific genes downloaded and mapped on the mouse genome. After checking expression level of Cd177 by clusters, a threshold of 1.5 was set to discriminate cells that are *Cd177* + (log-normalized expression value > 1.5) or *Cd177*- (log-normalized expression value < 1.5). Mean comparison *p*-values were calculated by means of stat_compare_means() function in ggpubr R package.

## Transposable Element quantification from RNAseq

To quantify the expression of Transposable Elements (TEs), we applied an analytic procedure that uses clustering in order to group individual TE copies that are supported by the same set of multimapping reads. This approach allowed us to exploit the information provided multimapping reads without having to discard them, at the cost of a lower resolution in terms of quantification of individual TE insertions. Reads were mapped to the mouse reference genome (GRCm38 assembly) with Star[84] using the outFilterMultimapNmax option set to 9999 to allow reporting all alignments without a limit on the number of mapping locations. The resulting BAM files for all samples were then concatenated and the coordinates of mapped reads were intersected (using bedtools intersect[85]) with those of TEs annotated in

RepeatMasker (v405). The resulting file was transformed with the tool awk to produce a binary matrix where columns correspond to each RepeatMasker TEs and rows to reads, and where the 0/1 value of each cell indicated whether the given read maps to the corresponding TE. This matrix was then transformed with mcxarray into a square matrix of Tanimoto distances between each pair of TEs and subjected to clustering using the Markov Cluster Algorithm[86] (MCL, filtering parameter >=0.1 Tanimoto distance. Inflation parameter 1.2). We then excluded clusters not containing any TEs of type LINE, SINE, or LTR, leaving in total 488.538 clusters that contain a variable number of TEs with common mappability profiles. The number of reads in each cluster was then quantified for each sample (discarding reads mapping to multiple clusters), and the resulting TE expression matrix was imported into R for differential expression analysis with DESeq2[87]. After selecting only high expression clusters (i.e. clusters with mean expression across all samples in the 1st quartile), cluster counts were normalized using size factors estimated from the number of reads uniquely mapping to Ensembl genes. We then performed differential expression analysis using the Wald Test and a design formula capturing the interaction between diet and treatment. The $p$-values thus obtained were then corrected for multiple hypothesis testing using the Benjamini-Hochberg procedure.

### DEPMAP analysis
Data were downloaded from DEPMAP (https://depmap.org/portal/interactive/) using the 22Q1 Chronos dataset for dependency and the Expression 22Q1 Public dataset for RNAseq. Regression analysis was performed in R version 3.6.2 with ggplot2 package.

### Statistics and reproducibility
No statistical method was used to predetermine sample size. No data were excluded from the analyses. The experiments were not randomized. The investigators were not blinded to allocation during experiments and outcome assessment.

For mouse survival, the Kaplan–Meier method with the logrank test to assess statistically significant differences in survival were used. In most cases, even when the groups were >2, we were not interested in all possible differences among all the groups but in few, specific pairwise comparisons, with a common control group as specified in the legends. In these cases, given the small number of simultaneous hypotheses, a simple Bonferroni correction was performed. Sample size was not based on prior specific hypotheses given the exploratory nature of the experiments. ELDA analysis was conducted on https://bioinf.wehi.edu.au/software/elda/.

For the in vitro molecular biological experiments, the biological replicas experiments mean independently cultured flask of cells. The replicate experiments are also available in the Source Data.

Statistical differences on continuous dependent variables were based on two-tailed t-tests, after assessment of distribution normality via the Shapiro test. In cases in which the independent categorical variable had more than 2 values, we applied 2-way ANOVA (with experiment as a block variable) and then applied Tukey's HSD test to identify pairwise statistically significant differences. Specific statistical tests for complex data are described in the relative method section.

All western blot was repeated at least thrice (unless otherwise indicated in the figure legend) from independent experiments with similar result.

### Reporting summary
Further information on research design is available in the Nature Portfolio Reporting Summary linked to this article.

## Data availability
The Whole Exome Sequencing, RNA-Seq, stranded RNA-Seq and scRNA-Seq generated in this study are deposited at The European Nucleotide Archive (ENA) under the accession number (PRJEB53822) and can be accessed publicly. Publicly available DEPMAP data (22Q1 Chronos dataset for dependency and the Expression 22Q1 Public dataset for RNAseq) were downloaded from the repository (https://depmap.org/portal/interactive/). All remaining data is available in the Article, Supplementary and Source Data files. Source data are provided with this paper.

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

## Acknowledgements

We acknowledge technical assistance from Marine Meliksetyan and Emanuele Bonetti (IEO, Milan, Italy). R.P. has been supported by a fellowship from the Fondazione Umberto Veronesi (FUV). This work was partially supported by core institutional funding through the Italian Ministry of Health Ricerca Corrente and 5×1000 funds, and by the following grants: AIRC IG 2017 (Project code 20162) to P.G.P, AIRC-Cariplo Foundation (TRIDEO 2014 n. 15650) to L.M., Italian Ministry of Health - Giovani Ricercatori (GR-2011-02350249) to L.M., European Hematology Association-Jose Carreras Young Investigator Award 2014 to L.M.

## Author contributions

Conceptualization: R.P., L.M., and P.G.P. Methodology: R.P., L.M., T.D., M.S., S.F., G.P., and P.F. performed the in vivo experiments; R.P, L.M., T.D., B.A.D., A.P., and D.T., performed in vitro experiments; R.P., S.P., A.X., D.V. and M.S. did breast PDX experiments; R.P and R.R contributed to generation of CFLAR overexpression, RNAseL known down and LSDKO cells; R.P. and S.R., FACS acquisition, sorting and analysis. E.G. performed RNAseq, scRNAseq and WES data analysis. T.L. performed ERVs analysis. M.E. contributed to initial in vivo experiments. Formal analysis: R.P., E.G., T.D., T.L., and L.M., Resources: L.M., M.G., M.V., E.C., L.L., S.M., and P.G.P. Data curation: R.P., L.M., E.G., and T.L. Writing and editing: R.P., L.M., and P.G.P. Supervision, P.G.P. Funding acquisition, P.G.P.

## Competing interests

R.P., L.M., T.D., S.M. and P.G.P. are listed as inventors of filed patent application PCT/EP2016/080156. The other authors declare no competing interests.

## Additional information

[1]Department of Experimental Oncology, IEO European Institute of Oncology IRCCS, Milan, Italy. [2]Center for Genomic Science of IIT@SEMM, Fondazione Istituto Italiano di Tecnologia, Milan, Italy. [3]Institute for Clinical Chemistry and Laboratory Medicine, University Hospital and Faculty of Medicine, Technische Universität Dresden, Dresden, Germany. [4]Medical Clinic I, University Hospital Carl Gustav Carus, Technische Universität Dresden, Dresden, Germany. [5]Mildred-Scheel Early Career Center, National Center for Tumor Diseases Dresden (NCT/UCC) University Hospital and Faculty of Medicine, Technische Universität Dresden, Dresden, Germany. [6]Cancer Cell Biology, Institute of Molecular Genetics of the Czech Academy of Sciences, Prague CZ-14220, Czech Republic. [7]IFOM ETS - The AIRC Institute of Molecular Oncology, Milan, Italy. [8]Department of Hemato-Oncology, Universita' Statale di Milano, Milan, Italy. [9]Department of Biomedical Sciences, University of Padova, Padova, Italy. [10]These authors contributed equally: Elena Gatti, Tiphanie Durfort. ✉e-mail: luca.mazzarella@ieo.it; piergiuseppe.pelicci@ieo.it

