## [Peer Review File · Nature Communications]

Reviewers' Comments:

Reviewer #1:

Remarks to the Author:

Pallavi et al provides a detailed characterization of the effects of calorie restriction (CR) on acute promyelocytic leukemia cells. Their studies suggest that CR shows an initial anti-tumor effect followed by re-emergence of disease via non-genetic selection of leukemia initiating cells associated with down-regulation of genes involved in dsRNA sensing and Interferon signaling. The CR phenotype was reversed by inhibition of LSD1, leading to re-activation of dsRNA/IFN signaling, enhanced apoptosis and leukemia eradication. The CR-LSD1 effects were mimicked by combining LSD1 ablation with insulin/IGF1 blockade, which sensitized cells to LSD1-induced death by inhibiting CFLAR. Synergy between CR and LSD1 inhibition was also seen in other AML types and in triple-negative breast cancer.

The authors have been responsive to the prior critique.

They have addressed the need to better characterize the LIC population and have done this based on cell surface phenotype and scRNA analysis and demonstrated increase in LIC with CR and reduction with LSD1 inhibition. They have also characterized differences in LIC between LIC and non-LIC populations and within LIC populations by flow.

They have also addressed the need to include healthy non-cancer controls and shown differential responsiveness to and further discussion of specific clinical context for potential translation.

Although validation of results in primary APL cells would be valuable, the authors do not feel that this is technically feasible at this time but do confirm results in primary MLL-AF9-bearing AML PDX and the mouse NPM1c+ model in vitro.

This work is interesting and significant since it describes the effects of CR in short-term tumor growth inhibition but survival of leukemia stem cells that support disease regeneration. It further elucidates the underlying mechanisms yielding drug combinations that may potentially be used to target the leukemia stem cell populations. The studies are carefully performed and limitations of specific analyses are acknowledged. The results will provide a basis for further studies to address as yet unanswered questions.

Review of Reviewer 2 comments:

Reviewer 2 felt that results of this work are valuable to increase knowledge regarding CR-mediated mechanisms for tumor growth modulation and cancer treatment. Experiments are well designed and controlled, and authors make a significant effort to demonstrate the different mechanistic cues involved in APL relapse using pharmacological and genetic approaches. However, the reviewer requested the following points to be addressed.

R2P1. - Authors show that after CR treatment, LICs are less prone to apoptosis (due to downregulation of the dsRNA sensing and IFN signaling), and initial anti-tumor effects are 'invariably followed by explosive disease and an expansion of cells with LIC activity'. It would be important for the authors to provide or generate data characterizing leukemic cells from CR-treated mice after phenotypic adaptation? Are LICs the main source of bulk leukemic cells during disease progression? If they specifically participate in the first phase of relapse, could inhibition of LSD1 in CR-treated mice have the same effectiveness at later time points of disease progression (after relapse)?

i) With respect to providing data characterizing leukemic cells from CR-treated mice after phenotypic adaptation, the authors have adequately addressed this point by adding RNAseq data of APL cells from CR-fed mice at 6 weeks post-injection. They observed significant inversion or attenuation of the main transcriptional features of the CR-treated phenotype observed at 4 weeks

at 6-weeks.

ii) Are LICs the main source of bulk leukemic cells during disease progression.

They have adequately addressed this point by mapping a LIC-enriched subpopulation in their scRNAseq datasets, featured by expression of CD34 and high levels of CD177. They show that LICs increase in frequency after CR and confirmed these data by prospective identification of LICs by FACS analyses, and in their scRNAseq datasets.

iii) could inhibition of LSD1 in CR-treated mice have the same effectiveness at later time points of disease progression (after relapse).

They contend that the speed of leukemia evolution in their model prevents evaluation of the effects of late treatments.

R2P2- An increase of double positive Irf8 and Gfi1 cells, which defines a rare phenotypical unstable progenitor prone to apoptosis, is predominantly observed in Cd177- cells upon LSD1 inhibition. However, LICs are characterized by Cd177 expression. If authors hypothesize that LICs are the main cell type involved in disease relapse after CR treatment, could they clarify why are Cd177- cells, which are not LICs, the ones in which they see a major effect after LSD1 inhibition?

The authors contend that transcriptional changes upon LSD1 inhibition should be considered transient and not compatible with cell survival and therefore the CD177+LIC population expresses Irf8 and Gfi1 at low levels. In this context, the significance of the non-LIC Irf8+ Gfi1+ population with regards to relapse and LSD1 inhibition effects needs clarification.

R2P3- Very low numbers of mice (n= 2-3) and in vitro replicates (N=2) in some experiments is a concern. A power calculation should be performed and provided to justify the experimental design with such a low number of replicates.

The explanations are reasonable

R2P4. Authors use two-tailed t-test throughout the study, even when comparing more than 2 groups (e.g., SD, CR, SD + LSD1i, CR + LSD1i). The data should be should be re-analyzed using the appropriate statistical analysis.

The argument to perform t-test rather than a multiple comparison test do not seem correct to me,

Reviewer #3:

Remarks to the Author:

The authors have reasonably addressed my concerns.

REVIEWERS' COMMENTS

Reviewer #1 - also asked to comment on R#2 (Remarks to the Author):

Pallavi et al provides a detailed characterization of the effects of calorie restriction (CR) on acute promyelocytic leukemia cells. Their studies suggest that CR shows an initial anti-tumor effect followed by re-emergence of disease via non-genetic selection of leukemia initiating cells associated with down-regulation of genes involved in dsRNA sensing and Interferon signaling. The CR phenotype was reversed by inhibition of LSD1, leading to re-activation of dsRNA/IFN signaling, enhanced apoptosis and leukemia eradication. The CR-LSD1 effects were mimicked by combining LSD1 ablation with insulin/IGF1 blockade, which sensitized cells to LSD1-induced death by inhibiting CFLAR. Synergy between CR and LSD1 inhibition was also seen in other AML types and in triple-negative breast cancer.

The authors have been responsive to the prior critique.

They have addressed the need to better characterize the LIC population and have done this based on cell surface phenotype and scRNA analysis and demonstrated increase in LIC with CR and reduction with LSD1 inhibition. They have also characterized differences in LIC between LIC and non-LIC populations and within LIC populations by flow.

They have also addressed the need to include healthy non-cancer controls and shown differential responsiveness to and further discussion of specific clinical context for potential translation.

Although validation of results in primary APL cells would be valuable, the authors do not feel that this is technically feasible at this time but do confirm results in primary MLL-AF9-bearing AML PDX and the mouse NPM1c+ model in vitro.

This work is interesting and significant since it describes the effects of CR in short-term tumor growth inhibition but survival of leukemia stem cells that support disease regeneration. It further elucidates the underlying mechanisms yielding drug combinations that may potentially be used to target the leukemia stem cell populations. The studies are carefully performed and limitations of specific analyses are acknowledged. The results will provide a basis for further studies to address as yet unanswered questions.

Response to Reviewer 1 comments: *We thank reviewer 1 for his valuable input on our manuscript. We thank the reviewer for acknowledging our effort to address his prior critiques successfully.*

Review of Reviewer 2 comments:

Reviewer 2 felt that results of this work are valuable to increase knowledge regarding CR-mediated mechanisms for tumor growth modulation and cancer treatment. Experiments are well designed and controlled, and authors make a significant effort to demonstrate the different mechanistic cues involved in APL relapse using pharmacological and genetic approaches. However, the reviewer requested the following points to be addressed.

R2P1. - Authors show that after CR treatment, LICs are less prone to apoptosis (due to downregulation of the dsRNA sensing and IFN signaling), and initial anti-tumor effects are 'invariably followed by explosive disease and an expansion of cells with LIC activity'. It would be important for the authors to provide or generate data characterizing leukemic cells from CR-treated mice after phenotypic adaptation? Are LICs the main source of bulk leukemic cells during disease progression? If they specifically participate in the first phase of relapse, could inhibition of LSD1 in CR-treated mice have the same effectiveness at later time points of disease progression (after relapse)?

i) With respect to providing data characterizing leukemic cells from CR-treated mice after phenotypic adaptation, the authors have adequately addressed this point by adding RNAseq data of APL cells from CR-fed mice at 6 weeks post-injection. They observed significant inversion or attenuation of the main transcriptional features of the CR-treated phenotype observed at 4 weeks at 6-weeks.

ii) Are LICs the main source of bulk leukemic cells during disease progression. They have adequately addressed this point by mapping a LIC-enriched subpopulation in their scRNAseq datasets, featured by expression of CD34 and high levels of CD177. They show that LICs increase in frequency after CR and confirmed these data by prospective identification of LICs by FACS analyses, and in their scRNAseq datasets.

iii) could inhibition of LSD1 in CR-treated mice have the same effectiveness at later time points of disease progression (after relapse).

They contend that the speed of leukemia evolution in their model prevents evaluation of the effects of late treatments.

Response to R2P1: *First, we would like to express our gratitude towards reviewer 1 for agreeing to review our responses to the reviewer's 2 comments. We are happy that with our additional experiments, we successfully answered the earlier concern of reviewer 2.*

R2P2- An increase of double positive Irf8 and Gfi1 cells, which defines a rare phenotypical unstable progenitor prone to apoptosis, is predominantly observed in Cd177⁻ cells upon LSD1 inhibition. However, LICs are characterized by Cd177 expression. If authors hypothesize that LICs are the main cell type involved in disease relapse after CR treatment, could they clarify why are Cd177⁻ cells, which are not LICs, the ones in which they see a major effect after LSD1 inhibition?

The authors contend that transcriptional changes upon LSD1 inhibition should be considered transient and not compatible with cell survival and therefore the CD177⁺LIC population expresses Irf8 and Gfi1 at low levels. In this context, the significance of the non-LIC Irf8⁺ Gfi1⁺ population with regards to relapse and LSD1 inhibition effects needs clarification.

Response to R2P1:

CD177⁻/Gfi1^{high}/Irf8^{high}: we contend that a double-high population, LIC or non LIC (or CD177⁻ or CD177⁺) is not compatible with cell survival. This cell type is not identified in normal bone marrow in the cited paper (ref 51), and in our system only appears upon LSD1 inhibition, which leads to massive cell death. We do not know if residual cells that relapse in some SD-LSD1i-treated mice remain double-high.

In either instance, we believe that a detailed discussion is not essential for the paper narrative, and we believe we sufficiently highlighted the partially speculative nature of these considerations by adding "presumably" in the sentence "became Gfi1^{high}/Irf8^{high}, a bi-differentiated cell state presumably not compatible with cell survival".

We will be happy to include these considerations to the manuscript if deemed necessary, although we fear this would require significant text and would unnecessarily interrupt the narrative.

R2P3- Very low numbers of mice (n= 2-3) and in vitro replicates (N=2) in some experiments is a concern. A power calculation should be performed and provided to justify the experimental design with such a low number of replicates.

The explanations are reasonable

Response to R2P3: *We thank reviewer 1 for accepting our explanations.*

R2P4. Authors use two-tailed t-test throughout the study, even when comparing more than 2 groups (e.g., SD, CR, SD + LSD1i, CR + LSD1i). The data should be should be re-analyzed using the appropriate statistical analysis.

The argument to perform t-test rather than a multiple comparison test do not seem correct to me,

Response to R2P4:

we repeated all statistical testing using ANOVA with post-hoc Tukey analyses to account for multiple hypotheses, even when we were not formally testing multiple hypotheses. All key results maintain statistical significance. These additional analyses are now also included in the Source data as well with the respective raw data information.

Reviewer #3 (Remarks to the Author):

The authors have reasonably addressed my concerns.

Response to Reviewer 3 comments: *We sincerely thank reviewer 3 for his insightful suggestions during the revision, and we are glad that we have successfully addressed all his concerns.*